# Heterogeneity and stochastic growth regulation of biliary epithelial cells dictate dynamic epithelial tissue remodeling

Kenji Kamimoto, Kota Kaneko, Cindy Yuet-Yin Kok, Hajime Okada, Atsushi Miyajima, Tohru Itoh*

Laboratory of Cell Growth and Differentiation, Institute of Molecular and Cellular Biosciences, The University of Tokyo, Tokyo, Japan

**Abstract** Dynamic remodeling of the intrahepatic biliary epithelial tissue plays key roles in liver regeneration, yet the cellular basis for this process remains unclear. We took an unbiased approach based on in vivo clonal labeling and tracking of biliary epithelial cells in the three-dimensional landscape, in combination with mathematical simulation, to understand their mode of proliferation in a mouse liver injury model where the nascent biliary structure formed in a tissue-intrinsic manner. An apparent heterogeneity among biliary epithelial cells was observed: whereas most of the responders that entered the cell cycle upon injury exhibited a limited and tapering growth potential, a select population continued to proliferate, making a major contribution in sustaining the biliary expansion. Our study has highlighted a unique mode of epithelial tissue dynamics, which depends not on a hierarchical system driven by fixated stem cells, but rather, on a stochastically maintained progenitor population with persistent proliferative activity.

*For correspondence: itohru@
iam.u-tokyo.ac.jp

**Competing interests:** The authors declare that no competing interests exist.

## Introduction

Tissue growth, maintenance, and remodeling play central roles in ensuring the structural and functional integrity of adult organs and are achieved through the coordinated actions of cell proliferation and differentiation. In these processes, the location, arrangement and timing of cell proliferation are tightly regulated in tissue-specific and context-dependent manners (*Barker et al., 2010*). Selected and dedicated populations of adult stem cells that continuously provide progenies to replenish aged or damaged cells can be found in some tissues, while multiplication of differentiated cells also plays an active role in normal tissue turnover and regeneration in other tissues. Elucidating the mechanisms that regulate the spatiotemporal pattern of cell proliferation in intact tissues will provide fundamental insights into organ homeostasis and regeneration.

The liver epithelium comprises the parenchymal cells, or hepatocytes, and the biliary epithelial cells (BECs), also known as cholangiocytes. In the adult liver, these cells are maintained in a quiescent state in which they stable epithelial sheets and tubules. They enter dynamic regeneration processes once the organ suffers tissue loss or various types of injury. In particular, the regeneration process that responds to chronic injury involves drastic changes in the morphology and phenotype of liver epithelial tissues, and in both human pathologies and animal models often accompanies a phenomenon called the ductular reaction (*Gouw et al., 2011*; *Michalopoulos and Khan, 2015*). The ductular reaction has been histologically characterized as the ectopic emergence and expansion of BEC-marker-positive cells in the liver parenchymal region. After using newly established imaging approaches to capture three-dimensional (3D) tissue morphology in situ, we recently reported that such a phenomenon results from dynamic and adaptive structural changes in intra-hepatic biliary tree architecture (*Kaneko et al., 2015*). Thus, ductular reaction essentially represents drastic and

**eLife digest** Cell proliferation – the process by which cells multiply – plays an important role in many biological processes, including tissue growth, maintenance and remodeling. In these processes, the way cells proliferate is reportedly related to their roles in the tissue and the structures that they form.

The biliary tree, a piping system that exists to drain the bile produced in the liver, forms a complex, tree-like, tubular structure. The biliary tree is essential for healthy livers to work well, and has been known to grow and change its structure quite dynamically during an injury or while the liver regenerates. However, it was not clear how biliary tree cells behave as the biliary tree grows and remodels itself. Does each cell behave in the same way? And how does cell growth relate to changes in the structure of the biliary tree?

Kamimoto et al. have now developed new methods to observe detailed three-dimensional tissue structures and to trace the behavior of single cells. Using these techniques to study a mouse model whose liver was injured by toxic chemicals revealed the behavior of biliary cells as they responded to the injury. None of the biliary cells proliferated uniformly, and there were some peculiar cells that proliferated quite vigorously compared to the others.

Kamimoto et al. then made a mathematical model that could explain cell behavior and tissue remodeling at different scales. This showed that the activity of those peculiar, rapidly proliferating cells was maintained by chance as the biliary tree expanded. These findings help us understand how the biliary tissue grows and the liver regenerates. They may also provide us with a clue to understanding the nature of the behavior of living things, which is sometimes seemingly ordered and robust, and sometimes unpredictable and mysterious.

It remains to be seen whether the new model can be applied to other types of tissues or in other species. Further work is also needed to investigate which genes and proteins are involved in controlling the behavior of cells in the growing biliary tissue.

complex remodeling of the biliary epithelial tissue, which is likely to be regulated by a sophisticated mechanism that controls BEC proliferation.

Notably, ductular reaction not only serves as an attractive model for studies of the mechanisms of tissue remodeling and cell proliferation, but also is a pathophysiologically relevant regenerative response of the liver to counter various types of injury stimuli (*Michalopoulos, 2014*). Several lines of evidence involving knock-out mice for regulatory signals, such as TWEAK, HGF/c-Met and FGF7, have collectively demonstrated that the suppression or failure of ductular reaction leads to exacerbated liver injury and severe defects in liver regeneration (*Ishikawa et al., 2012*; *Takase et al., 2013*; *Lu et al., 2015*). However, the in vivo behavior and manner of growth of BECs during biliary tree remodeling remains largely unknown. For many years, ductular reaction has been regarded and studied as a model that represents the activation of adult liver stem/progenitor cells, which may reside in the biliary tree and which can differentiate into hepatocytes or BECs (*Duncan et al., 2009*; *Miyajima et al., 2014*). Although in vitro studies have demonstrated the presence of clonogenic cells in the biliary compartment that are highly proliferative under culture conditions (*Miyajima et al., 2014*), the in vivo existence and behavior of such proliferative BEC subpopulations remain unclear. This was partly because many recent studies employing genetic-lineage tracing approaches in vivo have focused on the trans-differentiation capacities of BECs and of hepatocytes, rather than on the mode of proliferation of BECs themselves (*Grompe, 2014*; *Michalopoulos and Khan, 2015*)

Recently, single-cell approaches have been applied to the field of stem cell research (*Etzrodt et al., 2014*). Among these, in vivo quantitative single-cell tracing has successfully revealed the presence of stem/progenitor cell populations and their unique features in various organs, providing fundamental insights into the cellular basis of tissue homeostasis, regeneration and tumorigenesis (*Doupé et al., 2010*; *Driessens et al., 2012*; *Hara et al., 2014*). This technique is designed to reveal the rules of cellular dynamics that underlie tissue growth by tracking a population of single cells comprehensively and by deducing the characteristics of proliferative capacity, cell fate and

behavior through statistical analyses involving mathematical simulation. A clonal cell tracing study on BECs was recently reported (*Tarlow et al., 2014b*) that focused solely on the clonal differentiation potential of both hepatocyte and BEC lineages, but not on the clone size (representing the clonal growth potential). Thus, the exact cell numbers of BEC-derived clones were not quantified. The biliary tree has a highly complex and fine structure (*Kaneko et al., 2015*), thus it is practically impossible to count or even estimate the number of BECs that reside in it using conventional histological analysis in tissue sections. Hence, it was essential to develop a 3D imaging method that can provide detailed and reliable tissue structure images and thus allow quantitative assessment.

Here, we aimed to elucidate the basic mechanisms that underlie the morphological transformation of the biliary tree, a key process in liver regeneration. In order to study the precise cellular dynamics in the context of complex tissue structures with branching morphology, we introduced three new methods and strategies: high-resolution 3D imaging, quantitative single-cell tracing, and computational simulation. Using our newly established platform to visualize, label and trace BECs in liver tissue, we first revealed that the expansion and remodeling of the biliary tree in a mouse model of chronic liver injury was predominantly driven by the intrinsic growth of biliary epithelial tissue. We performed quantitative single-cell tracing of BECs in vivo to elucidate the underlying cellular behavior, which was further characterized by mathematical modeling and computational simulation. The results highlighted hitherto unrecognized heterogeneity among BECs and the mode of their proliferation, constituting the basis for the drastic structural transformation of the biliary epithelial tissue in regenerating liver in vivo.

## Results

### Ubiquitous and unbiased labeling of BECs using Prom1-CreERT2;R26R-tdTomato mice

In order to achieve specific and permanent labeling of BECs, we employed a mouse strain in which a tamoxifen-inducible variant of Cre (CreERT2) is knocked-in to the *Prominin1 (Prom1)* locus, hereafter referred to as the Prom1-CreERT2 mouse (*Zhu et al., 2009*). Prom1, better known as the surface antigen CD133, has been reported to be an 'oval cell/liver progenitor cell (LPC) marker' in injured liver (*Rountree et al., 2007*; *Dorrell et al., 2011*), but it is also expressed in BECs under normal conditions (*Suzuki et al., 2008*). We examined the expression pattern of the endogenous Prom1 gene product in the normal adult mouse liver, and confirmed that it was expressed in essentially the same manner as CK19 and EpCAM, which are robust and reliable markers of BECs (*Figure 1a,b*).

In the Prom1-CreERT2 strain, a nuclear localization signal-conjugated LacZ (nLacZ) gene was also knocked-in to the same Prom1 locus. X-gal staining experiments using liver sections showed that nLacZ was expressed in a BEC-specific manner (*Figure 1c*, left and right panels). We also performed whole-mount X-gal staining of the entire liver of the Prom1-CreERT2 mice. This resulted in 3D visualization of finely branching, tree-like architecture spreading throughout the organ (*Figure 1c*, middle and right panels), the pattern of which matches well with the biliary tree structure that we recently revealed using an ink-casting technique (*Kaneko et al., 2015*). The staining pattern was also consistent with 3D images of CK19 immunostaining (see below).

We crossed the Prom1-CreERT2 mice with a Cre-inducible fluorescent reporter mouse strain (R26R-tdTomato) for permanent labeling and tracing of BECs (*Madisen et al., 2010*). We first administered a relatively high dose of tamoxifen (10 mg/20 g mouse body weight) into Prom1-CreERT2; R26R-tdTomato mice and analyzed their livers to evaluate labeling specificity and efficiency. Immunostaining images (*Figure 1d*) and FACS plot (*Figure 1e* and *Figure 1—figure supplement 1*) showed that all of the labeled cells were included in the EpCAM$^+$ BEC population, indicating that the Prom1-CreERT2;R26R-tdTomato mouse can be used for specific labeling of BECs. The labeling efficiency at this dose of tamoxifen was approximately 30% (*Figure 1—figure supplement 1*). The labeling seemed to occur at random in the BEC population: the labeled tdTomato$^+$ cells were distributed ubiquitously among EpCAM$^+$ BECs with no apparent relation to features of the biliary structure, which will be described later in more detail. We also compared the Prom1-CreERT2 lineage-labeled and non-labeled cells in terms of their proliferative characteristics upon liver injury; using a 5-ethynyl-2'-deoxyuridine (EdU) incorporation assay, and found no significant difference between them (*Figure 1—figure supplement 2*), showing that their proliferative capacities are

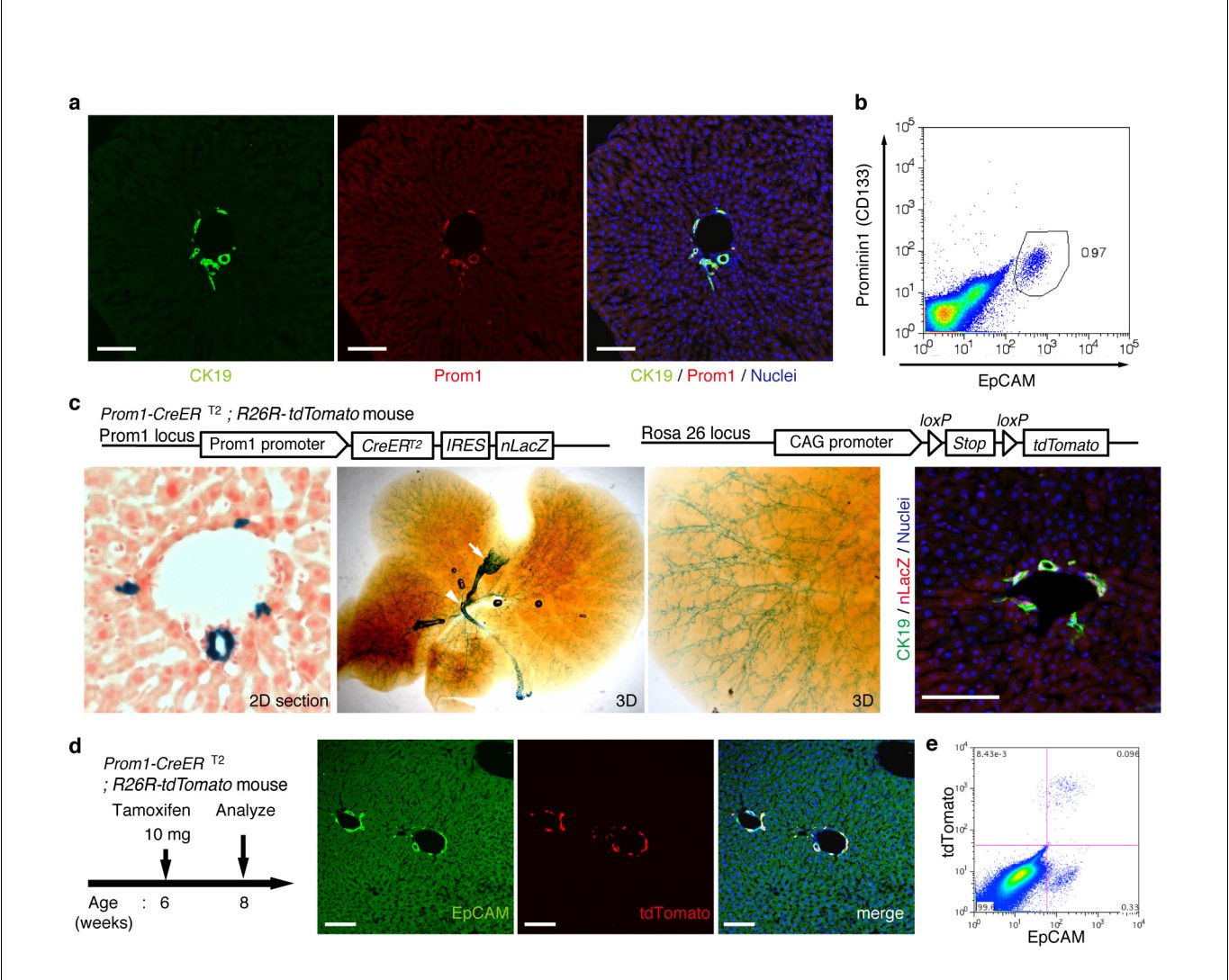

**Figure 1.** Visualization and lineage labeling of BECs using Prominin1 (Prom1) expression. (**a**) Immunofluorescent (IF) staining for CK19 and Prom1 in the adult mouse liver (scale bars, 100 μm). (**b**) Representative expression pattern of Prom1 and EpCAM by fluorescence-activated cell sorting (FACS) analysis. Each dot or point represents an individual cell. The colors (pseudo-colors) indicate the density of dots (i.e., cells), corresponding to increasing numbers of events from blue to red. The outlying (and outlined) group highlights the EpCAM[+] Prom1[+] double positive population, indicating that these markers are co-expressed with each other. Successive gates were applied for DAPI[-], forward and side scatter (FSC/ SSC), and pulse width (not shown). (**c**) Prom1-CreERT2;R26R-tdTomato mice were used to allow the detection of Prom1[+] cells on the basis of LacZ expression. X-gal staining was performed on tissue sections (left panel) and whole liver samples (middle panels). The latter was cleared with benzyl-alcohol and benzyl-benzoate (BABB) after staining. The intra-hepatic biliary tree, as well as the extra-hepatic bile duct (white arrowhead), and the base of the gallbladder (white arrow) were visualized. Liver sections were also stained with anti-LacZ and anti-CK19 antibodies (right panel). (**d** and **e**) Lineage labeling in the Prom1-CreERT2;R26R-tdTomato mouse liver after tamoxifen administration (scale bars, 100 μm). (**d**) Immunostaining of liver section. (**e**) Representative FACS plot pattern of labeled cells. Successive gates were applied for DAPI[-], FSC/SSC, pulse width and EpCAM[+] (not shown). All experiments were performed with at least four biological replicates.

The following figure supplements are available for figure 1:

**Figure supplement 1.** Quantification of lineage-labeled cells in Prom1-CreERT2; R26R-tdTomato mice.

**Figure supplement 2.** EdU uptake assay in lineage-labeled cells in Prom1-CreERT2;R26R-tdTomato mice.

**Figure supplement 3.** No labeled cells were detected in the absence of tamoxifen in Prom1-CreERT2;R26R-tdTomato mice.

indistinguishable. In addition, the BEC lineage labeling rate of around 30% did not change significantly, even after ductular reaction was induced by a chronic injury model (*Figure 1—figure supplement 1*). We also confirmed that no leakiness of labeling occurred in the absence of tamoxifen administration (*Figure 1—figure supplement 3*). These results indicate that the BEC labeling in our system occurs in an un-biased manner, and that the labeled cells faithfully represent the entire BEC population.

## 3D fluorescent imaging of the biliary tree with single-cell resolution

As shown in *Figure 1c*, the intrahepatic biliary epithelial tissue exhibits a complex yet ordered tree-like structure, which cannot be readily recognized by conventional histological analyses of thin tissue sections. We modified and improved a 3D immunostaining and imaging protocol using thick sections that we had recently reported (*Kaneko et al., 2015*), and established a new fluorescent 3D imaging platform with a single-cell resolution. As schematically depicted in *Figure 2a*, we cut liver samples into thick sections (200−500 μm thickness) and subjected them to immunostaining and optical clearing. The 3D image obtained by this method consistently recapitulated the basic structural unit of the biliary tree obtained by other protocols (*Kaneko et al., 2015*; *Takashima et al., 2015*) (*Figure 2b*). One or two duct tubes run alongside the portal vein (PV) (*Figure 2b*; PV is not shown), and many finer branches, which we call ductules hereafter, protrude from the duct and wrap around the PV (*Figure 2c*). This structural unit was observed around the entire biliary tree (*Figure 2—figure supplement 2a*).

## Pre-existing BECs make a major contribution to the TAA-induced biliary remodeling

We chose the thioacetamide (TAA)-induced chronic liver injury protocol as an experimental model system with which to study the dynamic morphological changes and cellular behavior of the biliary epithelial tissue. TAA is known to induce localized cell death of hepatocytes, specifically around the central vein (CV). Continuous administration of this drug causes chronic inflammation and bridging fibrosis, and eventually leads to tumor formation, reminiscent of the progression of fibrotic and cirrhotic liver disease in humans (*Yeh et al., 2004*; *De Minicis et al., 2013*). In addition to this pathophysiological relevance, pilot test experiments comparing several injury models also revealed that TAA caused the least autofluorescence in liver specimens (data not shown), which is suitable and advantageous for obtaining high-quality imaging data that can be used to perform analyses on detailed tissue and cellular structures.

Before proceeding with single-cell clonal tracing of BECs, we conducted tracing experiments in a condition where BECs were labeled en masse to analyze: the pattern of 3D changes in the morphology of the biliary structure; and the relative contribution of the pre-existing BECs to the expanded structures. Beginning 2 weeks after a single administration of a high dose of tamoxifen, the Prom1-CreERT2;R26R-tdTomato mice were continuously administered TAA (*Figure 2d*). We first confirmed the spatial distribution of the labeled cells under normal conditions at the onset of the injury protocol. As shown in *Figure 2e*, the BEC labeling occurred evenly in both the duct and ductule compartments, and in an un-biased mosaic pattern independent of other morphological features such as the duct size or branch locations.

We then analyzed the fate of labeled cells and branching morphogenesis over 8 weeks of the injury (*Figure 2f–h*). The biliary tree structure began to undergo dynamic remodeling by TAA 2 weeks, with ductules, which reside omni-directionally around the PV in the normal condition, extending uni-directionally toward the CV. The labeled tdTomato$^+$ cells were clearly observed in the extending ductular compartment, indicating that the pre-existing biliary epithelial tissue, presumably the ductule, has undergone the morphological change (*Figure 2f*, white arrows). At TAA 4 weeks, the biliary extension that reached deep into the liver lobule had begun to form intricately branched structures around the CV area. Many tdTomato$^+$ cells were observed in these newly formed branches. Remarkably, none of the expanded branches lost their connectivity with the main duct in the peri-portal area (*Figure 2g*, white arrows). This indicates that the ductular reaction is neither the migration of detached BECs nor ectopic emergence of BEC-like cells at a distant area in the liver parenchyma, but rather an extension of branching architecture from the pre-existing biliary tissue.

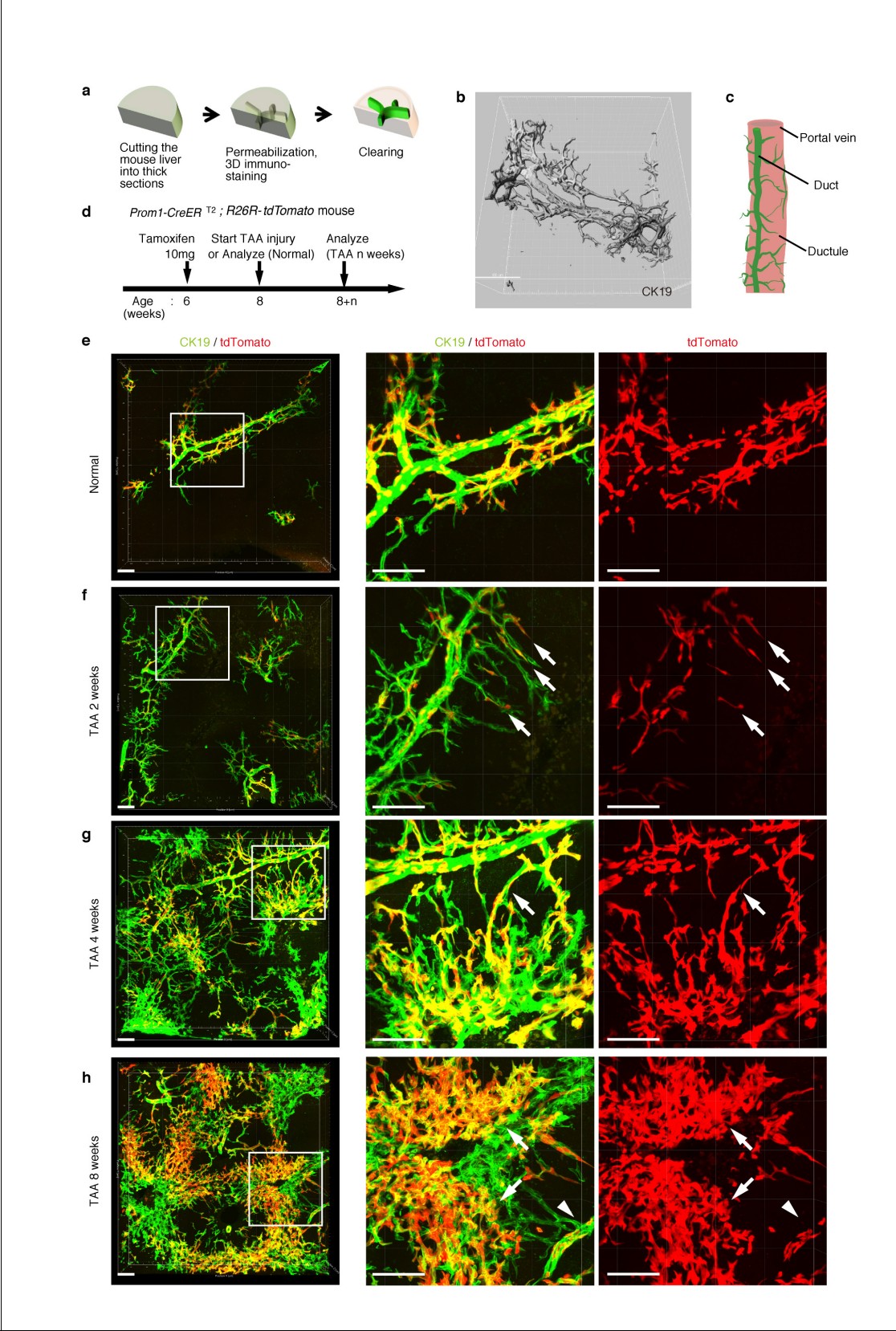

**Figure 2.** Pre-existing BECs contribute to the nascent biliary epithelial tissue structure upon injury. (a) Schematic illustration of the 3D imaging method used to observe biliary tree structures. (b) 3D imaging view of the normal biliary tree structure revealed by anti-CK19 immunostaining. Z-stacked images

*Figure 2 continued on next page*

*Figure 2 continued*
were acquired by confocal microscopy and reconstructed by IMARIS software (normal shading mode). (c) Schematic model for the biliary tree structure under the normal condition. (d) Experimental scheme. (e–h) 3D reconstructed images of the biliary tree revealed by anti-CK19 immunostaining (green), showing the distribution of the BEC lineage-labeled cells (red) in the expanded biliary structure. Serial z-stacked confocal images were tiled (3 x 3 tiles) automatically by automatic positioning stage and Olympus fluoview software. Data are displayed as maximum-intensity projections. A region indicated by a white box in the left panel is magnified in the middle and right panels. Scale bars represent 100 μm. (f) White arrows indicate that pre-existing BECs (tdTomato$^+$ cells) are extending outward. (g) White arrows indicate a branch of the biliary tree that connects the biliary duct around the PV with newly formed biliary branches around the CV. (h) White arrows indicate clusters of labeled cells that are located around the CV. White arrowheads indicate that the duct compartment around the PV shows a uniform mosaic pattern. All experiments were performed with at least five biological replicates.
The following figure supplements are available for figure 2:

**Figure supplement 1.** The level and distribution pattern of the ductular reaction in a microscopic view is highly diversified within a liver.
**Figure supplement 2.** Macroscopic view of the ductular reaction upon TAA injury over time.

Morphological changes further continued along with the disease progression, and at TAA 8 weeks, drastically expanded BECs appeared to make mesh-like network structures (*Figure 2h*).

We observed changes to the distribution pattern of the labeled BECs during the course of the injury progression. The labeled tdTomato$^+$ cells remained evenly distributed in a mosaic fashion in the main duct structure around the PV (*Figure 2h*, white arrowheads), but they displayed apparently uneven distribution in the newly formed structures. That is, the expanded compartment around the CV was composed of several clusters of tdTomato$^+$ cells (*Figure 2h*, white arrows), implicating clonal expansion of a BEC subpopulation therein.

Notably, we found two opposite features, diversity and uniformity, of the ductular reaction that were visible at the micro- and macro- scale , respectively. When subjected to microscopic observation, biliary tree within each defined region of interest exhibited diversity in terms of growth speed and structural features, particularly at its periphery, even in 3D images. Quantitative analysis of BEC distribution in tissue sections supported this notion (*Figure 2—figure supplement 1*). Taking advantage of the LacZ expression in BECs of the Prom1-CreERT2 mouse strain, we also examined the TAA-induced biliary tissue remodeling at the macroscopic scale (*Figure 2—figure supplement 2*). Organ-wide visualization of the entire biliary tree structure enabled us to grasp the landscape of the ductular reaction, in which whole-tissue remodeling follows an ordered and uniform pattern with respect to both spatial and temporal changes.

The results of the BEC lineage tracing in the 3D tissue architecture strongly suggested that pre-existing BECs made a major contribution to the ductular reaction in the TAA injury model, at least within the initial period of 8 weeks. It should be noted that, in the present experimental setting, virtually all of the labeled cells were contained within the CK19$^+$ cell population and that labeled hepatocytes were rarely detected (*Figure 2h*). This result is consistent with recent reports by many other groups who have used lineage tracing experiments, in that the cells that are induced and expanded by the ductular reaction, or LPCs, do not show stem or progenitor cell-like activity that contributes to new hepatocytes in most, if not all, models of mouse liver injury (*Grompe, 2014*; *Tarlow et al., 2014b*; *Yanger et al., 2014*).

## Hepatocytes are not the main source of the expanded biliary structure

On the basis of the results of the 3D tracing of BECs, we assumed that pre-existing BECs were the main source of the newly formed biliary structure. However, several studies have recently reported that hepatocytes are capable of converting to become BEC-like cells upon liver injury (*Michalopoulos et al., 2005*; *Sekiya and Suzuki, 2012*; *Yanger et al., 2013*; *Nagahama et al., 2014*; *Tanimizu et al., 2014*; *Tarlow et al., 2014a*). We thus sought to employ a complimentary lineage-tracing strategy to directly evaluate the contribution of hepatocytes as an alternative source of the expanded biliary structure upon TAA-induced ductular reaction.

For lineage tracing of hepatocytes, we used a recombinant adeno-associated virus vector pseudo-serotyped with capsid 8 (rAAV2/8), which expresses an improved version of the Cre recombinase

(iCre) gene under the control of a hepatocyte-specific promoter. The rAAV2/8 vector is well known to target hepatocytes in the mouse liver in a highly specific and efficient manner, and has been used in many studies to transfer genes into hepatocytes in vivo. Importantly, it does not transduce BECs in the adult mouse liver (*Yanger et al., 2014*). We injected rAAV2/8-iCre into the R26R-tdTomato mice to permanently label hepatocytes (*Figure 3a*). FACS analysis of liver cells isolated from these mice confirmed that almost all hepatocytes were labeled (more than 97%; *Figure 3b*), while EpCAM$^+$ BECs were not labeled (less than 0.1%; *Figure 3—figure supplement 1b*). We also confirmed the specificity and efficiency of the rAAV2/8-iCre-mediated labeling by using immunohistological analysis (*Figure 3c*).

Here, lineage-tracing of hepatocytes was performed in both the TAA and the DDC injury models, as the latter was used in recent reports to convincingly demonstrate the contribution of hepatocytes to the proliferating ductules (*Tanimizu et al., 2014*; *Tarlow et al., 2014a*). We analyzed the expression of several molecular markers that are associated with BECs/LPCs, including EpCAM, CK19, Prom1, Spp1 (also known as osteopontin), and MIC1-1C3. Upon DDC administration for 8 weeks, we detected many Spp1$^+$ cells in the tdTomato-labeled population, but we did not detect any EpCAM$^+$ tdTomato$^+$ cells in this population (*Figure 3d*). This indicates that while hepatocytes can indeed gain a 'biliary' phenotype in terms of Spp1 expression, they are still not fully converted to BECs as defined by EpCAM expression. FACS analysis using the MIC1-1C3 surface antigen instead of Spp1 also revealed that hepatocytes contributed to the MIC1-1C3$^+$ 'biliary' population but not to that of the EpCAM$^+$ subset (*Figure 3—figure supplement 1c*). We also examined the expression patterns of Prom1 and CK19 and found that they were essentially the same as that of EpCAM, but they were different from those of Spp1 and MIC1-1C3 (data not shown). These results are consistent with the report by Tarlow et. al. (*Tarlow et al., 2014a*) showing that Spp1$^+$ MIC1-1C3$^+$ proliferating ductular cells derived from hepatocytes upon DDC injury (which they named hepPDs) are distinct from those derived from pre-existing BECs (which they named bilPDs) with respect to the gene expression profile. Specifically, such hepatocyte-derived cells (hepPDs) show little or no expression of CK19, EpCAM, or Prom1. We also estimated the contribution of hepPDs to the biliary population using our BEC labeling system, in which a decrease in the labeling ratio of BECs would be expected if a large number of hepatocytes converted into BECs. We found no significant decrease of labeling ratio after DDC administration in EpCAM$^+$ cells. The ratio might be slightly decreased in MIC1-1C3$^+$ cells, but this remains to be confirmed as statistically significant (*Figure 3—figure supplement 2*). Considering the relatively small contribution of the hepatocyte-derived cells to the entire MIC1-1C3$^+$ population upon DDC injury (1.88% in *Figure 3—figure supplement 1*), the result overall is consistent with that of the hepatocyte-tracing experiments using the AAV system.

In the TAA model (as in the DDC model), we did not detect the emergence of hepatocyte-derived EpCAM$^+$ cells (*Figure 3e*). Intriguingly, even hepatocyte-derived Spp1$^+$ cells were not detected, suggesting that hepPDs are not induced under this injury condition. Consistently, FACS analysis detected no MIC1-1C3$^+$ cells or EpCAM$^+$ cells in the tdTomato$^+$ population (*Figure 3—figure supplement 1d*), further confirming the notion that hepatocytes do not convert into either BECs (or bilPDs) or BEC-like cells (or hepPDs) in the course of TAA-induced liver injury, even after 8 weeks. Again, the expression patterns of Prom1 and CK19 were the same as that of EpCAM, but different from those of Spp1 and MIC1-1C3 (data not shown). Thus, Spp1 and MIC1-1C3 mark a different and broader cell population from that defined by the expression of CK19, EpCAM and Prom1. We concluded that hepatocyte-derived duct-like cells (hepPDs) do emerge in the DDC model but not in the TAA model, and that they are distinguishable from bilPDs. This finding may be of pathophysiological relevance, in that the DDC model is a cholestatic liver injury model that primarily targets the biliary epithelial system, whereas the TAA model is a hepatotoxic injury model with less severe cholestatic disease phenotypes.

To gain an insight into the spatial relationship between hepPDs (Spp1$^+$ EpCAM$^-$) and the biliary tissue structure composed of bilPDs (Spp1$^+$ EpCAM$^+$ cells), we further applied the 3D immunostaining and imaging analyses. 3D images confirmed that hepPDs were barely detected under the normal physiological condition or upon TAA-induced injury. In the DDC model, however, hepPDs emerged around several parts of the pre-existing biliary architecture (*Figure 3f*). The tissue architecture composed of hepPDs was different from that composed of bilPDs: the former did not form any

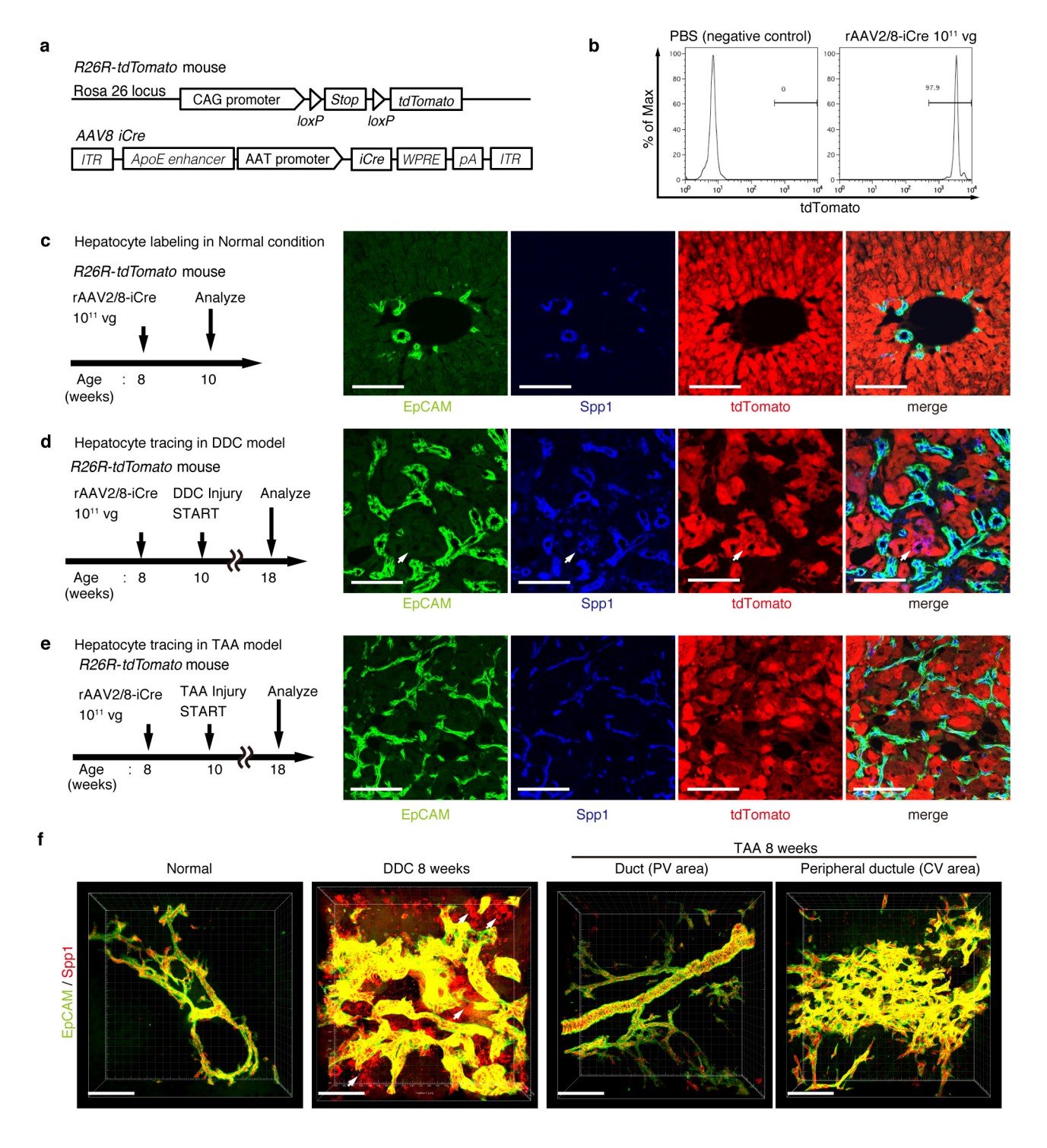

**Figure 3.** Lineage tracing of hepatocytes. (a) R26R-tdTomato mice were used in combination with rAAV2/8-iCre for the labeling of hepatocytes. rAAV2/8-iCre is designed to transduce only hepatocytes. (b) Representative image of FACS analysis of hepatocytes labeled by rAAV2/8-iCre. These histogram images show the result of serial purification gates (FSC/SSC, pulse width, DAPI-). (c) Adult R26R-tdTomato mice were injected with rAAV2/8-iCre ($1 \times 10^{11}$ vg /mouse). 2 weeks after injection, the mice were sacrificed and the livers were stained with anti-EpCAM and anti-Spp1 antibody (scale bar, 100 um). (d and e) Mice were injected with rAAV2/8-iCre ($10^{11}$ vg /mouse) and then subjected to a 3,5-diethoxycarbonyl-1,4-dihydrocollidine (DDC) or TAA injury model. tdTomato$^+$ Spp1$^+$ EpCAM$^-$ cells were only observed in DDC-fed mouse liver sections (white arrows). Analysis was done with 5 mice per each injury model. More than 6 sections were made per mouse. (f) 3D imaging was performed with WT mice (normal state, DDC for 8 weeks,

*Figure 3 continued on next page*

*Figure 3 continued*

TAA for 8 weeks). Acquired z-stack data is displayed as maximum-intensity projection after contrast adjustment with IMARIS software. In the DDC liver, Spp1$^+$ EpCAM$^-$ cells were observed (white arrow) around main biliary tubular structures that were composed of EpCAM$^+$ cells.

The following figure supplements are available for figure 3:

**Figure supplement 1.** FACS analysis of hepatocyte-derived cells in the TAA and DDC models.

**Figure supplement 2.** FACS quantification of labeled ratio in the Prom1-CreERT2; R26R-tdTomato mice before/after DDC injury.

single and contiguous branch structure but rather formed a subsidiary structure, located just around the pre-existing branched structures of bilPDs.

The results of these fate-tracing and 3D-imaging analyses confirmed that although hepatocytes are capable of being converted into the biliary epithelial state in a context-dependent manner, the main contributors to the ductular reaction (in terms of gene expression, cell number, and 3D structure) are pre-existing BECs. These findings also indicate that we can focus solely on the fate and behavior of pre-existing BECs when attempting to determine the cellular basis of the ductular reaction, particularly when using the TAA model. We have to note that this study does not refute the existence of hepatocyte-to-biliary phenotypic conversion, nor its contribution to liver regeneration. We wish to point out that hepPDs and bilPDs are different and their significance should be discussed separately.

## BECs exhibit widespread heterogeneity in proliferative capacity in situ

Based upon the notion that the nascent biliary structure is formed upon TAA injury through intrinsic cell growth in the biliary epithelial tissue, we set out to perform quantitative single-BEC tracing experiments in vivo in order to elucidate the cellular basis of this dynamic tissue remodeling process (*Figure 4a,b*). Prom1-CreERT2;R26R-tdTomato mice were administered a very low dosage of tamoxifen to label BECs at a very low frequency. 3D imaging and FACS analyses showed that the labeling was introduced at a rate of less than 0.2% of total BECs, and that the labeled cells located singly and apart from each other (*Figure 4c,d*). After single-cell labeling, mice were subjected to the TAA protocol and the size of single-cell-derived clonal progenies were analyzed during the course of injury progression. Of note, the labeling index of 0.2% did not change even after the injury period, which is consistent with the notion that the labeling was introduced in an unbiased manner (*Figure 4—figure supplement 1*). After 6 weeks of injury, there were still many BECs that had rarely divided and that remained as single cells or merely as clusters composed of a few cells. At the same time, we also observed larger colonies composed of dozens of cells, indicating that some BECs had undergone several rounds of cell division (*Figure 4e,f*). This clearly indicates that the BEC population does not proliferate uniformly as a whole upon injury, but rather, BECs exhibit heterogeneity with regard to their proliferative capacity in vivo.

Taking advantage of the 3D imaging, we quantified the exact size (i.e., the numbers of constituent cells) of each single-cell-derived clone in the liver tissue. We classified the biliary epithelial tissue into two distinct portions according to their tube diameter sizes and locations, namely 'duct' and 'peripheral ductule', as we anticipated a putative relationship between the tissue architecture and cell proliferation capacity. This highlighted two important features of colony size distribution. First, the duct compartment contained no large colonies of more than five cells, while the peripheral ductule compartment did contain large colonies (*Figure 4e–g*). Notably, we found no large colonies that were directly connected to the duct cells adjacent to PV. Thus, it is not likely that some proliferative cells reside in a fixed position in the duct compartment and continuously supply progeny toward the periphery. Rather, the colony-initiating, proliferative cells may reside in the peripheral ductule compartment and relocate their position upon parenchymal injury.

The second feature of colony size distribution relates to the pattern if this distribution. We initially assumed that the distribution of colony size would have a bimodal shape, with one peak corresponding to a non-proliferative population and the other to a proliferative one. However, the empirical data showed a unimodal distribution pattern with long tail (*Figure 4g and i*). Although the

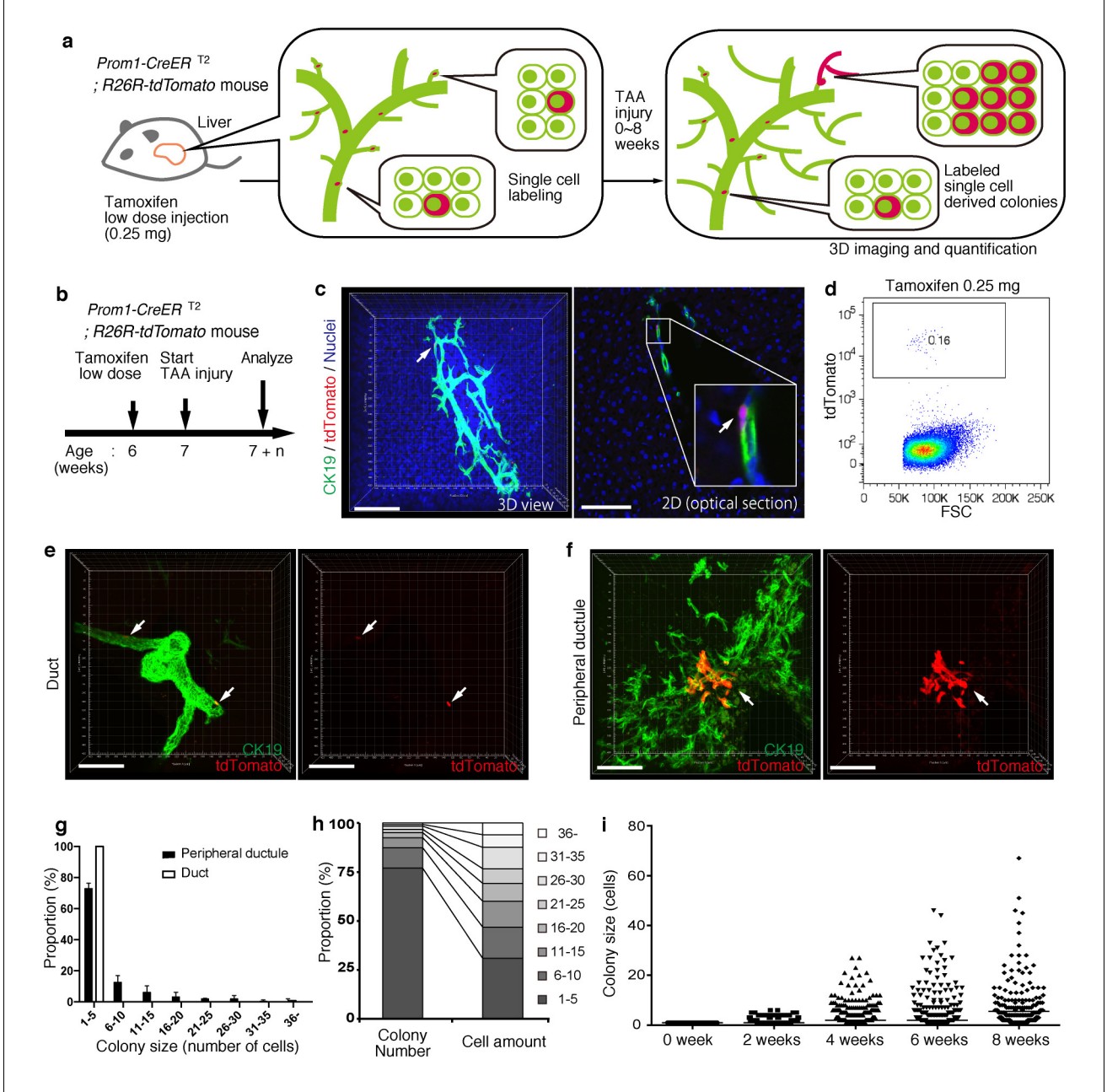

**Figure 4.** BECs exhibit heterogeneity in terms of proliferative capacity in vivo. (**a**) Schematic diagram showing the rationale for quantitative in vivo single-BEC tracing. (**b**) Experimental design. (**c**) Upon administration of a very low dosage of tamoxifen (0.25 mg/kg body weight), liver samples were stained with anti-CK19 antibody and Hoechst33342. BEC labeling was introduced at the single-cell level (white arrows). A 3D image and an optical section corresponding to the same visual field are shown in the left and right panels, respectively (scale bars, 100 µm). (**d**) Quantification of the BEC-labeling efficiency after the low-dosage tamoxifen injection. For FACS analysis, successive gates were applied for DAPI[-], FSC/SSC, pulse width and EpCAM[+] (not shown). A representative plot pattern for 4 biological replicates is shown. (**e** and **f**) 3D images of labeled colonies after 6 weeks of TAA injury. Thick sections were stained with anti-CK19 antibody and 3D images were acquired with tdTomato[+] colonies (white arrows) using confocal microscopy. The data are shown as maximum intensity projections. (**e**) Duct compartment around the PV area. (**f**) Peripheral ductule compartment around the CV area (scale bars, 100 µm). (**g**) Distribution of the quantified colony size at TAA 6 weeks (n = 5 mice, mean ± SD). The colonies were classified into two categories (duct and peripheral ductule) as described in the 'Materials and methods' section. (**h**) Relative numbers of colonies categorized by colony size as depicted in the legend to the right (left stacked bar chart), and the relative contribution of cell amounts from each colony category (right stacked bar chart) (calculated as follows: 100 x (sum of the cell numbers in a colony size)/(sum of all the counted cell numbers)). (**i**) Scatter plot of the colony size distribution over time. Data from five mice were pooled for each time point (total colony numbers counted were 257, 272,

*Figure 4 continued on next page*

*Figure 4 continued*

304, 307 and 310 for the 0, 2, 4, 6 and 8 week samples, respectively). Horizontal lines show the mean of colony size. Images shown in panels (c), (e), and (f) are representative data for at least 5 biological replicates.
The following figure supplement is available for figure 4:

**Figure supplement 1.** Labeling ratio in the Prom1-CreERT2; R26R-tdTomato mice was not changed after 8 weeks of TAA injury.

proportion of large colonies was seemingly low, these colonies actually incorporated a relatively large proportion of the total number of proliferated cells (*Figure 4h*). The results thus highlight a proliferative and expandable subpopulation of BECs that should make a major contribution to the growth of biliary epithelial tissue.

## Proliferating BECs are not strictly compartmentalized in tissue structure

We further sought to reveal the relationship between cell proliferation and tissue structure in detail. In many tissues, proliferating cells, or stem cells, are spatially arranged within a specific area that has a characteristic tissue structure; for example, intestinal stem cells reside in the crypt bottom of the intestinal epithelium (*Barker et al., 2010*). We searched 3D images of the biliary tree for any structural feature corresponding to the location of cycling cells by immunostaining for the cell-proliferation marker Ki67. Most of the Ki67$^+$ BECs localized to the peripheral area of the biliary tree in the TAA injured liver, whereas only a few Ki67$^+$ BECs were seen in the duct around the PV area (*Figure 5a* and *Figure 5—figure supplement 1*). This distribution pattern fits well with the results of the clonal cell tracing experiments. In the peripheral ductule compartment, the Ki67$^+$ cells were widely scattered. No further sign of a defined stem cell niche, such as a cluster or aligned arrangement of Ki67$^+$ cells, was observed. At 2 weeks of TAA administration, Ki67$^+$ BECs were already enriched in the peripheral ductule region, and scattered therein, rather than in the ducts (*Figure 5b*). This suggests that the mode of proliferation is maintained over time. We also examined the distribution pattern of proliferating BECs by continuous labeling of cycling cells using 5-bromo-2'-deoxyuridine (BrdU) incorporation, and obtained the same result (*Figure 5c*).

## BECs can take two different states based on their temporal proliferative capacities

The quantitative single-cell tracing data reveal heterogeneity among BECs with regard to their proliferative capacity in vivo. To understand the mechanistic basis for such heterogeneity, we sought to construct a simple growth model that could potentially explain and be used to simulate the proliferative behavior of BECs upon injury. For this purpose, it was necessary to collect information about cellular events in a short time scale. We hypothesized that BECs could be classified into two states based on their temporal proliferative capacities, and if this was the case, then the relationship between these states could be examined.

We designed an experiment to investigate the temporal proliferative state and cell fates of BECs using two nucleotide analogs, BrdU and EdU, like that employed in a previous study on pancreatic progenitor cells (*Teta et al., 2007*). First, mice were administered with BrdU continuously for 8 days to label in vivo as many proliferating cells as possible (1$^{st}$ label). After a short interval, mice were then administered the other nucleotide analog, EdU, for pulse labeling (2$^{nd}$ label) (*Figure 6a*). The rationale for this consecutive and double labeling experiment is as follows. If BECs can be classified into two distinct states: an actively and continuously dividing state and a quiescent state, then those BECs that are subjected to the 2$^{nd}$ labeling (corresponding to the actively and continuously dividing cells) should also have undergone the 1$^{st}$ labeling and thus are mostly observed as the 1$^{st}$ and 2$^{nd}$ label double-positive population. Conversely, if BECs are able to convert between the proliferative and quiescent states reversibly, they can divide at arbitrary time points and hence the 1$^{st}$ and 2$^{nd}$ labelings will occur independently. In this case, significant numbers of the 2$^{nd}$ label single-positive cells should be observed (*Figure 6b*). This experimental strategy can thus allow us to reveal the mode of cell proliferation in terms of the cells' potential for short-term transition between the

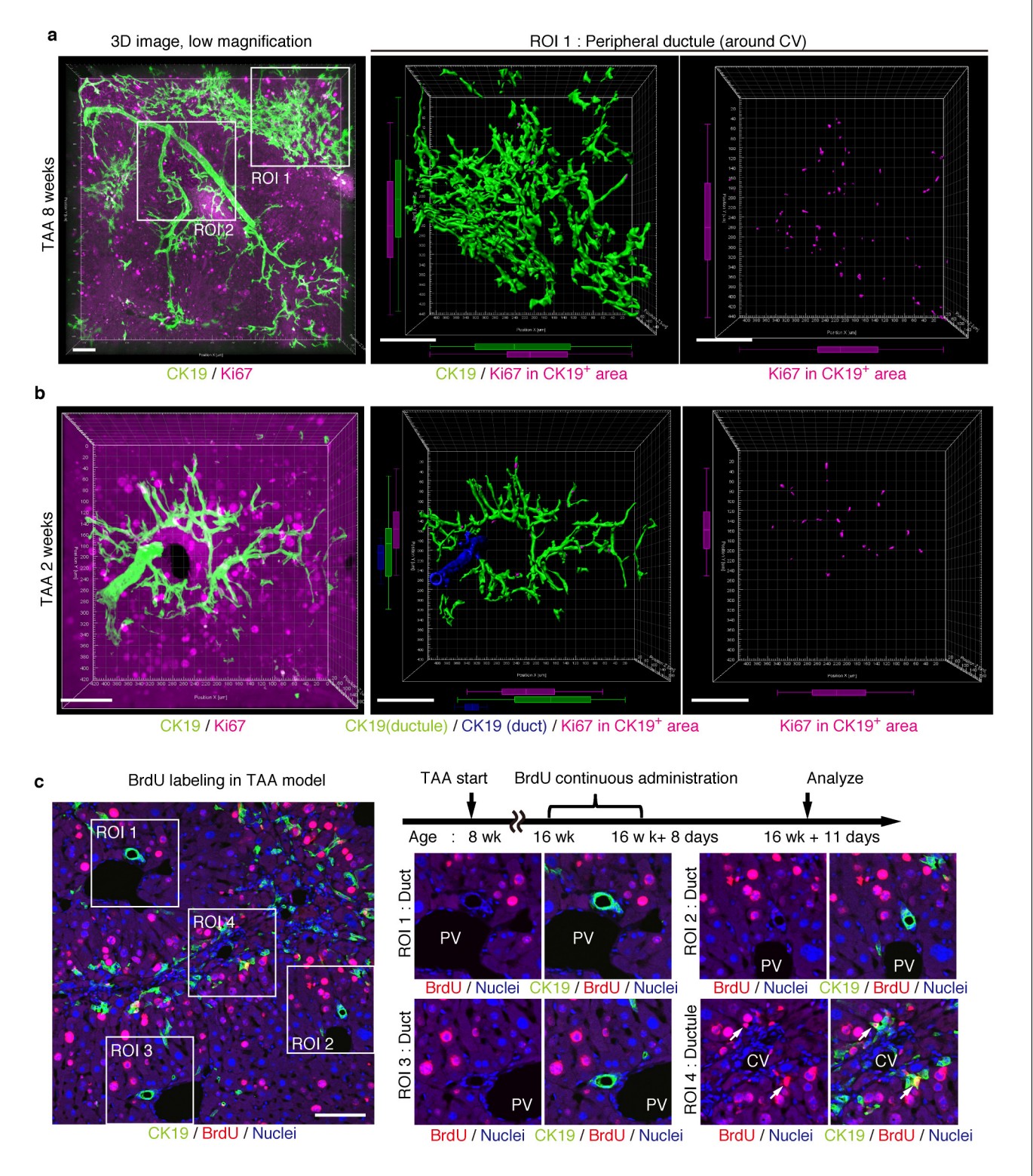

**Figure 5.** Proliferating BECs are scattered in the peripheral branching architectures in the biliary tree. (**a**) 3D images of the biliary tree (CK19 immunostaining; green) and the cell cycle marker Ki67 (magenta) in TAA-injured liver samples. Middle and right panels show magnified views of the region of interest (ROI) 1 shown in the left panel, where CK19+ area and Ki67+ BECs therein were extracted using the IMARIS surface protocol. Distribution patterns of the CK19+ area and the Ki67+ CK19+ cells were calculated using the IMARIS vantage protocol after the signals were projected onto the background, and depicted in 2D box-and-whisker plots. (**b**) Liver samples at TAA 2 weeks were analyzed as in (a). Biliary structure is classified

*Figure 5 continued on next page*

*Figure 5 continued*
into duct compartment (shown in blue in the center image) and ductule (green). (**c**) Proliferating cells were labeled by continuous administration of BrdU for 8 days in the course of the TAA injury and were analyzed by anti-BrdU immunostaining (magenta). BrdU incorporation was observed in BECs residing in the peripheral ductule compartment (white arrows), but rarely in those in the duct compartment. All experiments were performed with at least 3 biological replicates.
The following figure supplement is available for figure 5:

**Figure supplement 1.** Distribution of Ki67$^+$ BECs in the duct unit.

proliferative and quiescent states. We must note that the definition of proliferative capacity in this particular set of experiments is based on relatively short-term cell behavior, and the proliferative or quiescent state of a current cell does not apply to its progeny. In other words, BECs are classified by their transient cell division manner, but they are not restricted to this state permanently.

We observed many BrdU and EdU double-positive cells in the biliary tissue structure, and these cells constituted the major part of the EdU-labeled BEC population (*Figure 6c,d*). By contrast, among non-BEC (CK19$^-$) populations, the majority of the EdU$^+$ cells were BrdU$^-$, confirming that the presence of the double-positive cells was not attributable to insufficient washout of the 1$^{st}$ label. We also compared the proportion of BECs that were double-positive in our experiment with that theoretically predicted on the assumption that BECs proliferate in an unregulated manner (*Figure 6b*, lower panel), and found that their difference was statistically significant (*Figure 6e*). These results suggest that the proliferation pattern of BECs conforms to a model in which BECs can take two different states and change from one to the other irreversibly (*Figure 6b*, upper panel).

## A stochastic growth state change model explains the proliferation dynamics of BECs in vivo

By combining our data from quantitative single-cell tracing (*Figure 4*) with that on short-term proliferation behavior (*Figure 6*), we finally sought to establish a mathematical model that can explain the dynamics of biliary epithelial tissue growth in vivo. In the process of fitting a putative tissue growth model, we employed computational simulation by Markov chain Monte Carlo methods using R software to obtain numerous outputs for different sizes of BEC colonies, which eventually converged into a specific distribution pattern. Many rounds of simulation, validation, and modification of different parameters in various combinations were performed until we obtained a simulation result that fitted well with the empirical data from clonal tracing in vivo. (See 'Materials and methods' for details).

Considering the results of our nucleotide double-labeling experiments (*Figure 6*), we assumed a simple growth model driven by two cell states, continuously proliferating cells and quiescent cells. As stated before, we recognized that the most distinctive feature of the quantitative clonal tracing data was the heterogeneity in BECs with regard to proliferative activity, as manifested by the unimodality and long-tailed pattern of the colony size distribution (*Figure 4g and i*). To construct a model that recapitulates this feature, we hypothesized that there must be a factor that accounts for the heterogeneous proliferative capacities of each cell. Thus, we incorporated a stochastic cell behavior into our two-state growth model, in that a cell in the proliferative state can alter its growth state to the quiescent state in a probabilistic manner (*Figure 7a*). We use the term 'stochastic' here in exactly the same manner as in previous studies (*Doupé et al., 2010*; *Driessens et al., 2012*): to describe that the state of each cell is unpredictable rather than being fixed or pre-determined, with the total proportion of cells in each divergent state being balanced. It is important to note that the concept of stochastic growth does not mean that the transition process occurs in a non-regulated manner. There are indeed factors that regulate cellular growth state and behavior, although the switching and maintenance of each cell's state, timing of each cell division, and duration of each cell cycle 'appear' to be stochastic. Such stochastic behavior has already been shown to exist in progenitor cells in the interfolliculer epidermis (IFE) (*Doupé et al., 2010*; *Driessens et al., 2012*), and we defined parameters on the basis of that IFE model (*Figure 7a*). As expected, the model with the stochastic feature produced a widespread distribution in colony size, with each single proliferating cell yielding colonies of various sizes along the time course (*Figure 7b*).

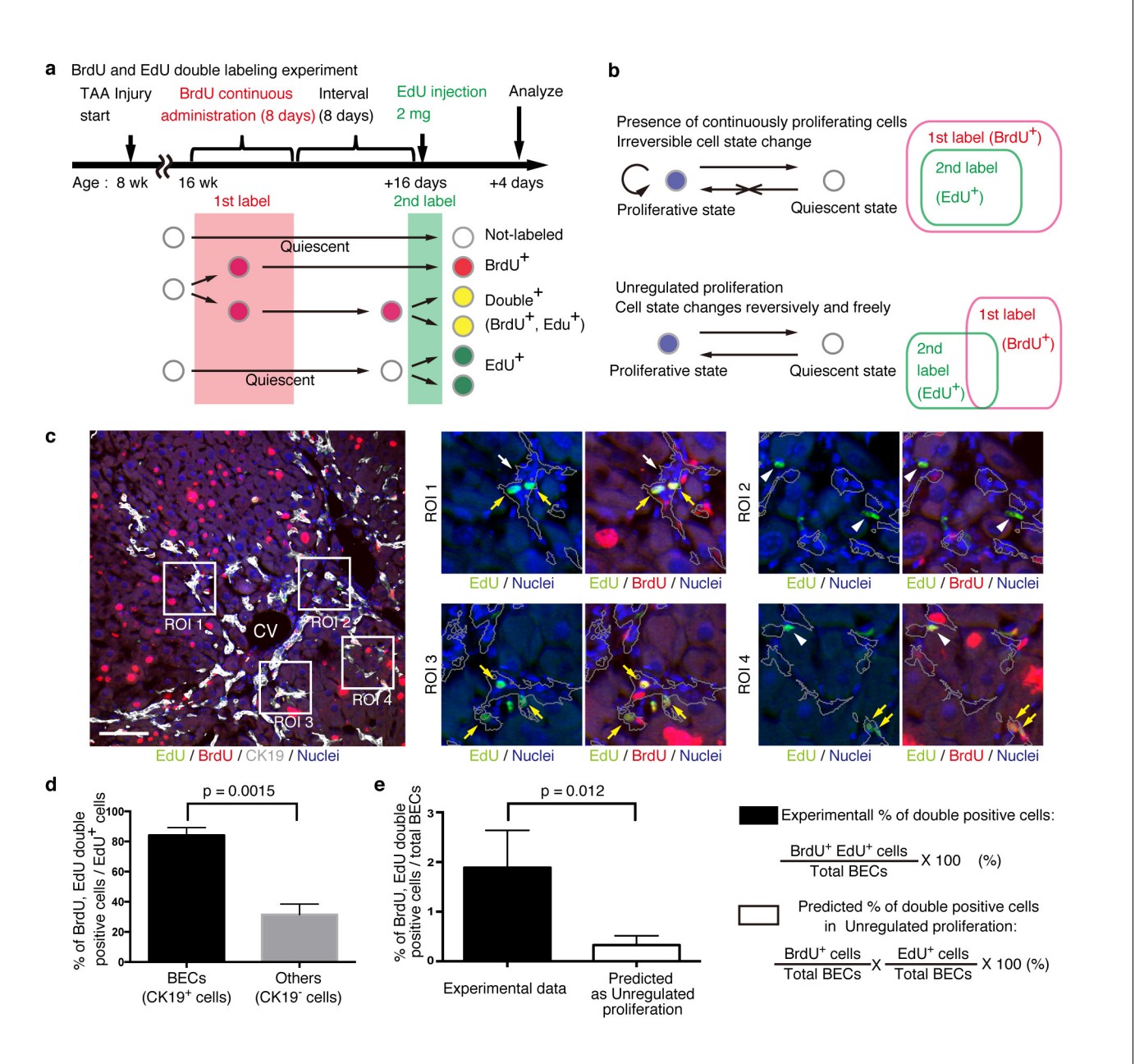

**Figure 6.** BECs do not proliferate uniformly and can be subdivided into those in the proliferative state and those in the quiescent state. (**a**) Schematic diagram for the experimental design. At week 8 of the TAA injury model, mice were given BrdU via drinking water (0.8 mg/ml) for 8 days (1st label). After an interval of a further 8 days, the mice were intraperitoneally injected with EdU (2 mg/20 g body weight) for pulse labeling (2nd label). (**b**) Schematic diagram depicting two possible growth modes. In the top model, proliferative and quiescent cell populations can be distinguished by temporal state. This growth mode will give an experimental result in which the 2nd label+ cells are included within the 1st label+ cells. In the second model (bottom), the cells change their growth state in an unregulated manner, resulting in an un-biased labeling pattern. (**c**) Immunofluorescent staining results for BrdU and EdU incorporation together with CK19 immunostaining. Representative regions of interest (ROI 1–4) in the left panel are shown in the middle and right panels. In the magnified images, the boundaries of the CK19+ areas are delineated in gray lines. White arrowheads, EdU+ BrdU- cells; white arrows, non-labeled cells; yellow arrows, BrdU+ EdU+double-positive cells. Scale bar represents 100 μm. (**d**) Quantification of the BrdU+ EdU+ double-labeled cells. Data represents the mean ± SEM for 5 mice. (**e**) Comparison of the incidence of the BrdU+ EdU+ cells between the experimentally obtained data and the result predicted by the assumption that the BrdU and EdU labelings occur independently. p-value was calculated by two-tailed paired Student's t-test. All experiments were performed with 5 biological replicates.

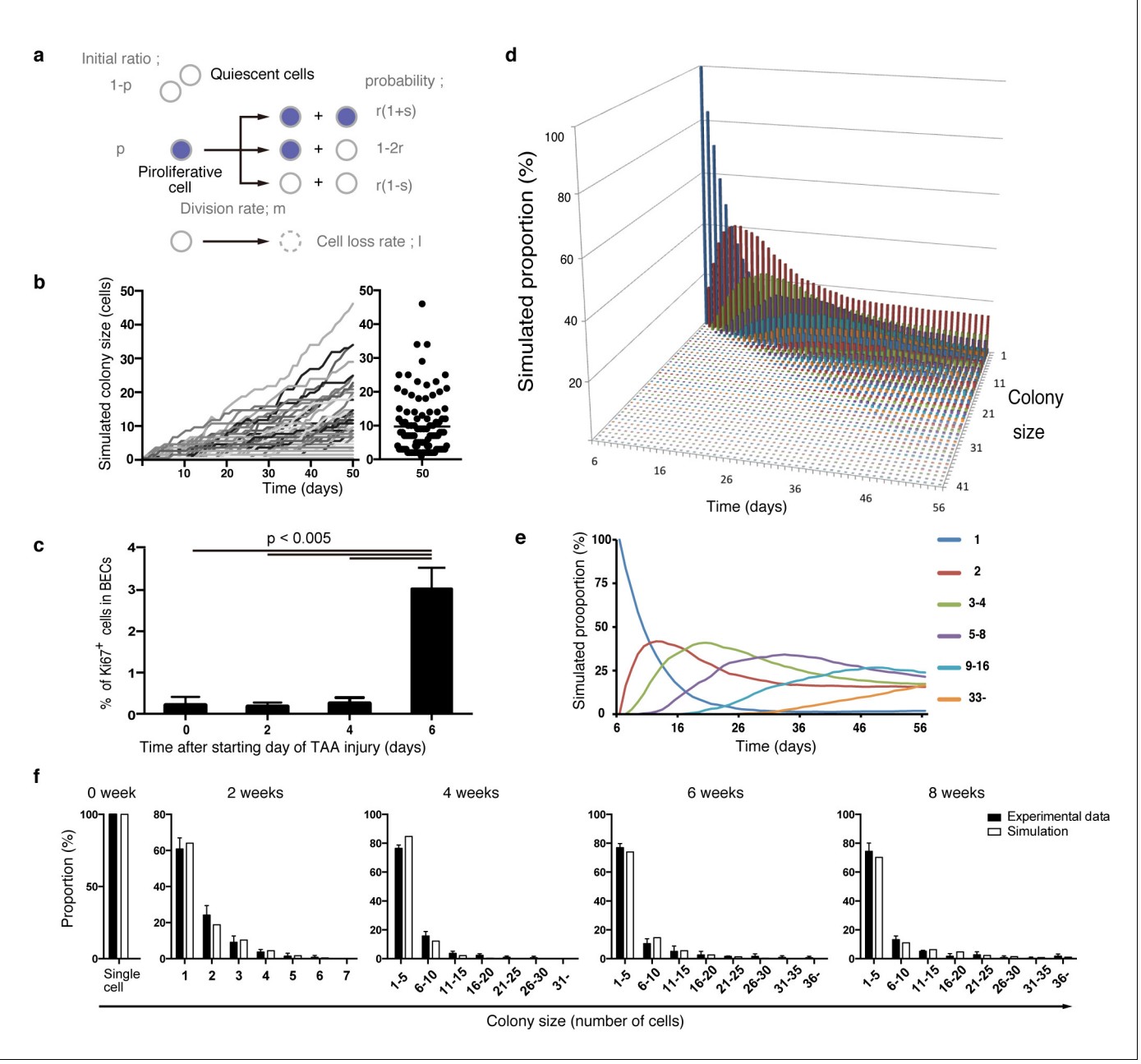

**Figure 7.** Mathematical modeling predicts the stochastic nature of the BEC proliferation. (**a**) Schematic diagram of the proposed growth model. We set five parameters. The parameter p represents the initial ratio of the cells in the proliferative state compared to total cells; m represents the probability that a cell will enter the cell cycle within one day. The frequency of each cell fate is defined by two parameters, r and s: r affects stability of colony size, whereas s represents imbalance of cell fate selection between proliferative and quiescent states. l represents the probability that a cell will be lost due to cell death (expressed as lost cells per total number of cells per day). (**b–e**) Monte Carlo simulation was performed using R software. (**b**) Consecutive changes in colony size over time are shown in the graph on the left. Each line represents a colony derived from a single proliferative cell. The scatter plot on the right corresponds to the colony-size distribution after 50 days of growth. (**c**) Quantitative data for Ki67[+] BECs upon TAA injury revealed by immunostaining. Day 0 samples correspond to livers under normal conditions. A significant increase in Ki67[+] cells among the BEC population was observed between 4 and 6 days after the start of injury (p<0.005, two-tailed paired Student's t-test). Data represent mean ± SEM for 4 mice. (**d** and **e**) A simulated pattern for distribution of colony size and changes in this distribution for colonies derived from a single proliferative cell. Results are shown for a simulation when the parameters were set as follows: m = 0.175, r = 0.15, s = 0.06, and l = 0.001 (simulated cell number = 2000). (**f**) The quantified in vivo data alongside simulation results. The data simulating changes in colony size were constructed from the data for a single proliferative cell (**d** and **e**) compensated for the presence of quiescent cells (the initial ratio p=0.465). The experimental data were derived from the data sets used in *Figure 4i*.

*Figure 7 continued on next page*

*Figure 7 continued*

The following figure supplements are available for figure 7:

**Figure supplement 1.** Discordance observed during the model-fitting process before taking account of the timing of the start of BEC proliferation upon injury.

**Figure supplement 2.** TAA causes death of hepatocytes around CV, but not death of BECs.

During the model-fitting process, the overall simulation result almost adequately fitted the experimental in vivo clonal tracing data, but there was an inevitable discordance, specifically at the time point of 2-week injury (*Figure 7—figure supplement 1*). A simulation result for 8-day injury period fitted significantly better with the experimental data for 2-week tracing than did simulated data for 2 weeks of injury, implying a gap of several days between the timing of colony formation in vivo and in silico. The initial simulation model was based on the assumption that the BECs began to enter the proliferative mode on the day when the injury was induced. We reasoned that the gap might be due to an incorrect estimation of the timing of the commencement of BEC proliferation upon injury, as there must be a lag between the timing of the injury application and that of the concomitant induction of tissue growth. To test this hypothesis, we performed additional immunostaining experiments using Ki67 and found that BECs first entered the cell cycle in vivo 6 days after the TAA injury was started (*Figure 7c*). This result was consistent with our prediction, and we modified the parameter for the starting point in our revised simulation model accordingly. This modification allowed the obtained patterns of simulation to fit the experimentally obtained in vivo data completely over the time course examined (*Figure 7d–f*).

We further evaluated our model in an additional experiment. Our model predicted that the rate of 'cell loss' should be minimal in the BEC population (model parameter l = 0.001, which means that only 0.1% of BECs would be lost in a day). We evaluated the occurrence of cell death in TAA injury by an in vivo cell-death detection method (*Edwards et al., 2007*). As expected, almost no cell death was detected in the CK19$^+$ area (*Figure 7—figure supplement 2*), thus supporting our model prediction.

Taken together, our results strongly suggest that the proliferation dynamics of BECs upon TAA injury conform to the following stochastic model: (i) upon liver injury, some BECs are activated to change their growth state to the proliferative state, while the others remain quiescent; (ii) those BECs that are in the proliferative state can stochastically and irreversibly convert back to the quiescent state during the course of injury; (iii) the highly proliferative BEC subpopulation is stochastically maintained and produces a large number of progenies, thereby making a major contribution to the expansion and remodeling of biliary epithelial tissue.

## Discussion

Understanding the relationship between the mode of cell proliferation and the resultant structural organization within a tissue is one of the fundamental issues in stem cell biology. Here, we sought to reveal this relationship by focusing on the ductular reaction, a unique and dynamic remodeling of the intrahepatic biliary tree that occurs upon chronic liver injury. Our results have delineated a progressing morphological transformation of the peripheral ductular structure in the TAA model, which is primarily achieved by a tissue-intrinsic cell expansion within the pre-existing biliary epithelium. Intriguingly, the expansion of biliary epithelial tissue is dictated by heterogeneous and stochastic behavior of BECs. Thus, a subset of proliferating BECs that is stochastically maintained, rather than a pre-determined and hard-wired progenitor population, exhibits clonogenic growth potential in vivo and plays a central role in the remodeling of biliary epithelial tissue.

In the present and recent studies, we have shown that the biliary epithelial tissue comprises of clearly distinct structural units (*Figure 2b,c*) (*Kaneko et al., 2015*). In many instances, the architecture of the biliary tree is illustrated as a simple monolayer tube with a luminal structure that corresponds to the ducts, with little or no attention being paid to the presence of peripheral ductules. However, it is the ductule compartment, rather than the ducts, that makes a key contribution to tissue remodeling, as exemplified by the fact that the majority of the proliferating BECs

reside in the peripheral ductule region (*Figures 4g*, *5* and *Figure 5—figure supplement 1*). Structural and functional units found in many different organs, such as the intestinal crypt and the hair follicle, exhibit fairly ordered structure in that they are reiteratively aligned in quasi-two-dimensional sheet-like tissue structures. By contrast, the biliary tree, especially the ductule compartment, spreads omni-directionally and it is quite difficult to capture its structure comprehensively and reproducibly. Therefore, it is necessary to use 3D-imaging methods to capture the tissue structures and cellular growth dynamics therein accurately and precisely. In order to observe detailed features and connectivity within the biliary tree structure in the liver, we have established a novel 3D staining and imaging method. There are substantial gains to be made by adapting the method so that it can be performed easily and effectively with minimum time and cost, especially for experiments such as quantitative single cell tracing analysis in which large numbers of samples must be handled. We have thoroughly optimized our 3D imaging protocol at various steps to render it easy and cost-effective; our protocol does not require expensive reagents or specialized equipment, and thus provides a versatile method for studying various tissue structures in the liver, as well as in other organs.

Using quantitative single-cell tracing, we revealed cellular heterogeneity in the biliary epithelial tissue in vivo. More specifically, we found two levels of heterogeneity among the BEC population. First, the biliary epithelial tissue can be subdivided into two classes, namely the duct and peripheral ductule compartments, on the basis of morphological and geographical characteristics. At the population level, the BECs in the peripheral ductule compartment have higher proliferative tendency than those in the ducts (*Figures 4g*, *5* and *Figure 5—figure supplement 1*). Second, among the BECs in the peripheral ductule compartment, there exists further heterogeneity in terms of the proliferative behavior. Not all of the ductule cells exhibit a proliferative response upon injury, and the size of the colonies varies even among the proliferative population. Both of these levels of heterogeneity can be explained by our mathematical model. The model predicts the initial ratio of the proliferating BEC population to be p=0.465, meaning that the majority of BECs do not divide in response to the injury stimulus. This is consistent with the notion that the duct compartment cells, which constitute the major part of the BEC population under the normal condition, do not or rarely divide upon injury (*Figure 5*). The proliferative heterogeneity in the peripheral ductule compartment is well represented in the model as change in stochastic growth state (i.e., proliferating vs. non-proliferating states) of BECs.

The dual nucleotide labeling experiment showed that the proliferative state of BECs switched irreversibly to a quiescent state (*Figure 6*). This feature has also been reported in a previous study of epithelial tissue in vivo (*Doupé et al., 2010*; *Driessens et al., 2012*), but is not consistent with a report that described the manner of cell division and cell fate decision in an in vitro model (*Spencer et al., 2013*). We think that this discrepancy may be due to the role of BECs in vivo. During the remodeling and growth process in biliary epithelial tissue, BECs must form a functional tubular structure. We assume that BECs may gain a type of steady-state phenotype (or, become more matured and differentiated cells) in order to generate and maintain a functional and robust epithelial tubular structure. From this point of view, it is reasonable to assume that their proliferation state changes irreversibly in vivo. Unfortunately, we do not yet have an established marker that can clearly distinguish BECs in different stages in the course of their functional differentiation and maturation, which prevented us from directly evaluating this possibility. Nevertheless, it could be supported in part by our findings that revealed the relationship between the proliferative capacity of BECs and structural features of the biliary tree (*Figure 4g*), as BECs constituting different tissue structures may be in distinct differentiation stages.

The opposing concepts of 'stochastic' versus 'deterministic' cell-fate decisions have been established as an important paradigm in stem-cell biology (*O'Neill and Schaffer, 2004*; *Enver et al., 2009*). It should be noted that the concept of stochastic versus deterministic regulation, which relates to *Figures 4* and *7*, does not refute the existence of temporal proliferative sub-states, as shown in the dual nucleotide labeling experiment (*Figure 6*). Here, the deterministic model refers to restriction of a cell to a state that is permanently fixed. In both models, cells are divided into subpopulations according to their transient proliferative state, rather than according to their permanent state. Thus, the existence of a proliferative state does not refute the stochastic model, and the existence of a temporal proliferative state does not simply mean that cell proliferation follows the deterministic model. Nevertheless, the mechanisms that govern the maintenance and regulation of cells are defined differently in the stochastic and deterministic models. In the stochastic

model, tissue stem cells, for example, have non-uniform cell fate and the consequences of cell fate selection are not predictable, whereas in the deterministic model, the stem cells give rise to their progeny uniformly and stably. The stochastic growth model first emerged in the 1960s, in order to explain the heterogeneous behavior of transplanted hematopoietic stem cells (*Siminovitch et al., 1963*; *Till et al., 1964*). Since then, many studies have revealed the existence and role of stochastic behaviour in various phenomena in living things (*Samoilov et al., 2006*). In our simulation, we used a stochastic feature to model not only the cell-fate decision process but also the regulation of the cell cycle length (*Figure 7*). In our mathematical simulation, we defined quiescent cells as those that do not divide at all, thus we do not need to consider their cell cycle length. We did, however, have to consider the cell cycle length of proliferative cells. We first ran many simulations with a uniform cell cycle length for all cells. We found, however, that the best fit was achieved when we allowed variation in cell-cycle length among the cells. This variation was defined as the parameter m (see 'Materials and methods' for details). Interestingly, this result agrees with the notion proposed by Spencer et al. (*Spencer et al., 2013*), confirmed by their single-cell experimental system, that there is substantial cell-to-cell variation in cell cycle lengths and delay between mitogen re-stimulation and CDK activation. They also reported that cells had heterogeneous inter-mitotic times, ranging from 20 hr to >50 hr. This fits well with our stochastic proliferation model whose definitions include varying inter-mitotic times.

In the stochastic model, the number and state of stem cells are maintained unstably. This fits the structural features of the biliary tree well in many regards. First, as the injury progressed, the biliary tree formed numerous arborization and branch ends around the injured area (*Figure 2e–h* and *Figure 2—figure supplement 2*). In addition, the location of cycling cells appeared to be scattered at the 3D level (*Figure 5a*). Such frequent remodeling and sporadic proliferation are well suited to the stochastic growth model, which can maintain a spatiotemporally flexible population of proliferating cells. Second, the stochastic model is consistent with diversity in tissue structure at the micro scale. In microscopic view, the stochastic system shows a degree of instability because the cell state may fluctuate and the output may be unpredictable. This feature nicely explains the dynamics of biliary remodeling, in that the detailed shape and degree of expansion of the extended biliary tree varied in each small area of interest (*Figure 2h*). This is why the architecture of the biliary tree looks complicated when observed by 2D liver section. Thus, the instability shown in the stochastic model matches the microscopic structural diversity of the biliary tree. Third, we focus on a macroscopic feature. In our stochastic model, each cell behaves in a flexible manner, but the total percentages of cells that take each cell-fate decision is strictly defined and follows a particular value at the population level (*Figure 7a*). This feature of the stochastic model makes the simulation outcome converge into an ordered pattern (*Figure 7c*). In other words, the stochastic growth mode provides a stable, robust and consistent outcome in a macroscopic view. This macroscopic feature corresponds to the biliary tree architecture in the macroscopic view, as the biliary tree appears to follow a uniform pattern in both the spatial and temporal terms when it is observed at a large scale (*Figure 2—figure supplement 2*).

In conclusion, the stochastic behavior of BECs plays a fundamental role in establishing a flexible and adaptive tissue remodeling system in the biliary epithelium, which underlies the robust regenerative capacity of the liver upon injury. Studies in other systems, including hematopoietic stem cells and epidermal homeostasis, have revealed similar models and concepts (*Siminovitch et al., 1963*; *Doupé et al., 2010*), implying that a common and fundamental mechanism governs stochastic cell behavior. Knowledge on the mode of tissue growth provided by this study applies to systems beyond the hepatobiliary system, and should provide significant insights into tissue dynamics and homeostasis in many other organs.

## Materials and methods

### Mouse strains and animal experiments

All animal experiments were conducted in accordance with the Guideline for the Care and Use of Laboratory Animals of The University of Tokyo, under the approval of the Institutional Animal Care and Use Committee of the Institute of Molecular and Cellular Biosciences, The University of Tokyo (approval numbers 2501, 2501–1, 2609 and 2706). Prom1-CreERT2 knock-in mice (*Zhu et al., 2009*)

and R26R-tdTomato reporter mice (*Madisen et al., 2010*) were purchased from Jackson Laboratory and maintained on a C57BL/6J background. Wild-type C57BL/6J mice were purchased from CLEA Japan (Japan). 6- to 10-week-old mice, including both males and females, were used unless otherwise specified.

For liver injury models, mice were administered with 0.3% (wt/vol) TAA in drinking water or fed a diet containing 0.1% 3,5-diethoxycarbonyl-1,4-dihydrocollidine (DDC). Tamoxifen was dissolved in corn oil and administered via oral gavage. In labeling experiments with nucleotide analogs, BrdU was administered via drinking water (0.8 mg/ml) and EdU for pulse labeling via intra-peritoneal injection.

For the hepatocyte labeling experiment, rAAV2/8-iCre was packaged in HEK293 cells according to the protocol described previously (*Kok et al., 2013*). The iCe-expressing plasmid vector was constructed by replacing the GFP insert in the pAM-LSP1-eGFP vector with the iCre gene. iCre was taken from pDIRE, which was a gift from Rolf Zeller (Addgene plasmid # 26745, [*Osterwalder et al., 2010*]). The titered rAAV2/8-iCre was injected by intraperitoneal injection ($1 \times 10^{11}$ vector genome / mice).

## Flow cytometric analysis

Liver cell preparation and FACS analyses were done as previously described (*Okabe et al., 2009*). Non-parenchymal cells were obtained from the mouse liver by the two-step collagenase liver digestion method and used for FACS analysis. Mice were laparotomized under isoflurane inhalation anesthesia, and 15 ml of pre-warmed (at 37°C) Liver Perfusion Medium (17701–0381–038, Life technologies, Waltham, MA) was perfused through the liver from the portal vein. The liver was then perfused with 20 ml of collagenase solution containing collagenase type IV (C5138-5G, Sigma Aldrich, St. Louis, MO) and fetal bovine serum (FBS, S1820-500, Biowest, France). The liver was harvested and placed into Dulbecco's Modified Eagle Medium (D5796, Life technologies) after removing the gallbladder and extra-hepatic bile duct, and minced gently with surgical knives. Roughly digested liver samples were filtered through a cell strainer (70 μm). The remaining undigested clots were further treated with digestion solution containing collagenase type IV, DNase I (DN25-5G, Sigma Aldrich), and pronase (10165921001, Roche, Switzerland). After removing the parenchymal fraction (hepatocytes) by centrifugation, the non-parenchymal fraction in the supernatant was treated with hemolysis buffer to remove red blood cells. The remaining cells were incubated with anti-FcR antibody, followed by staining with the antibodies listed in *Table 1*. Stained cells were suspended in phosphate-buffered saline (PBS) containing 3% FBS and 4',6-diamidino-2-phenylindole (DAPI, D3571, Life technologies) for dead cell staining.

**Table 1.** List of antibodies used in this study.

| Antibody | Company/Source | Host animal | Method | Dilution |
|---|---|---|---|---|
| Prominin1/CD133 (APC-conjugated) | Biolegend (San Diego, CA) | Rat | FACS | 1:100 |
| Prominin1/CD133 (purified) | eBioscience (Santa Clara, CA) | Rat | IF | 1:100 |
| EpCAM (FITC-conjugated) | (*Okabe et al., 2009*) | Rat | FACS | 1:200 |
| EpCAM (purified) | BD Pharmingen (Franklin Lakes, NJ) | Rat | IF | 1:200 |
| CD45 (APC-conjugated or APC-Cy7-conjugated) | Biolegend | Rat | FACS | 1:200 |
| MIC1-1C3 | STEMGENT (Lexington, MA) | Rat | FACS | 1:200 |
| Spp1 | R&D systems (Minneapolis, MN) | Goat | IF | 1:200 |
| Cytokeratin 19 | (*Tanimizu et al., 2003*) | Rabbit | IF | 1:200 |
| Ki67 | eBioscience | Rat | IF | 1:200 |
| LacZ | Abcam (United Kingdom) | Chicken | IF | 1:200 |
| BrdU | Abcam | Rat | IF | 1:200 |
| Hnf4a | Santa Cruz (Dallas, TX) | Goat | IF | 1:200 |

In the EdU uptake experiments, we used the Click-iT Plus EdU Alexa Fluor 647 Flow Cytometry Assay Kit (C10634, Life technologies) and Fixable Viability Stain 450 (562247, BD Biosciences, Franklin Lakes, NJ) according to the manufacturers' instructions.

Data were acquired using a FACSCanto II cell analyzer (BD Biosciences) or the MoFlo XDP (Beckman Coulter, Brea, CA). Final data analyses were performed with FlowJo software.

## Conventional (2D) immunofluorescent staining of thin liver sections

Whole-liver samples from adult mice were fixed with paraformaldehyde (PFA) as follows. The liver was perfused via the portal vein sequentially with 10 ml of PBS, 10 ml of 2% PFA, and 10 ml of 4% PFA. Upon harvest, the liver was cut into several blocks, placed into a 15 ml tube containing 4% PFA, and incubated for 12 hr. After fixation, the liquid was changed to 20% sucrose in PBS. The liver blocks were embedded in Tissue-Tek O.C.T. compound (Sakura Finetek, Japan) and snap frozen. Frozen samples were cut into 10-μm thickness using a cryostat-microtome (HM525, Microm International, Germany) and stained with the antibodies described in *Table 1*. For BrdU immunodetection, sections were heat-treated in TE buffer in an autoclave (120°C, 5 min) for antigen retrieval. For EdU detection, sections were treated with Click iT Plus EdU Alexa Fluor 488 Imaging Kit (C10637, Life technologies). Nuclei were counterstained with Hoechst33342 (H1399, Life technologies).

## 3D fluorescent imaging of mouse liver tissues

The mouse liver samples were pre-fixed using the protocol described for the conventional 2D staining. Note that the fixation by perfusion steps were essential to keep the tissue structure intact and to prevent soluble proteins from leaking out. The frozen liver samples were cut into thick sections (200–500 μm) using the cryostat-microtome. At this point, the samples set on the stage of the microtome were briefly warmed (by touching with gloved fingers) immediately before sectioning to prevent the samples from cracking. Samples were placed in PBS in disposable tubes, washed with PBS twice more, and then incubated with blocking/permeabilization reagent (3% FBS, 0.02% sodium azide, and 0.2% Triton X-100 in PBS) for 30 min at room temperature (RT). The same blocking/permeabilization reagent was also used in the following staining process for antibody dilution. We typically stained 10 thick sections in a 2 ml tube with 500 μl of diluted antibody solution. The samples were incubated in the antibody solution on a rocking or shaking device at 4°C for 2 days. The tubes were inverted by hand once a day to facilitate thorough mixing and uniform staining. The samples were washed with PBS twice and then transferred into a new 50 ml tube with 40 ml of PBS to wash out excess primary antibodies by incubating on a rocking device at 4°C for 2 days. After thorough washing, samples were treated with fluorescence-conjugated secondary antibodies using the same process as that used for primary antibody staining. Nucleotide staining dye, such as Hoechst33342, was mixed into the secondary antibody solution if necessary. Samples were then washed thoroughly with PBS again. After staining and washing, samples were treated with the tissue-clearing reagent SeeDB (*Ke et al., 2013*). The SeeDB reagent contains fructose as the main component and is safe, inexpensive, and easy to handle. We compared various different types of clearing reagents and obtained the best results in liver tissue imaging with SeeDB.

To perform quantitative single-cell tracing analyses based on tdTomato fluorescence, we stained liver cell nuclei by treating sectioned samples with SYTOX Green (in PBS containing 0.1% Triton X-100) on a rocking device overnight at RT. The samples were washed once with PBS and then directly placed into SeeDB.

For data acquisition, we used confocal microscopes (FV1000 or FV1200, Olympus, Japan) with a 30x silicone immersion lens (UPLSAPO30XS, Olympus).

We classified BECs into two compartments, duct and ductule, based on two criteria. First, location within the biliary tree structure was considered. In injured liver, the duct still runs alongside the PV, while the ductule is expanded to the outer parenchymal area. Second, we measured the diameter of the internal luminal space of the biliary structure. Biliary epithelial tubular structures with a luminal diameter of more than 8 μm were defined as the duct compartment, while the other tubular structures with smaller diameter were defined as the ductule.

## 3D X-gal staining of whole-liver samples

For organ-wide 3D visualization of the biliary tree, we performed X-gal staining of whole-liver samples derived from the Prom1-CreERT2 mice in which nLacZ is also knocked-in to the *Prom1* locus. The liver was perfused with 10 ml of ice-cold PBS containing 2 mM $MgCl_2$, then with 10 ml of fixative solution (0.2% PFA, 0.1 M HEPES, 2 mM $MgCl_2$, 5 mM EGTA, pH 7.3). The liver was incubated with fixative solution for 48 hr at 4°C with a daily change of the solution. The fixed liver was then treated with detergent buffer (0.1 M phosphate buffer, pH 7.3, 2 mM $MgCl_2$, 0.01% sodium deoxycholate, 0.02% Nonidet p-40) for 24 hr at 4°C. Next, the liver was treated with staining buffer (1 mg/ml X-gal in detergent buffer) for 48 hr at 4°C (from this step on, sample tubes were wrapped with foil for shading) and further for 12 hr at 37°C. At all incubation steps, samples were put on a rocking device. After washing out staining buffer with PBS, the liver was dehydrated with ethanol and then cleared with a 2:1 benzyl benzoate:benzyl alcohol (BABB) solution.

## In vivo cell death detection

For evaluation of cell death in injured liver, we performed a cell death detection assay (*Edwards et al., 2007*) with some modification. 200 µl of EthD-3 (0.2 mg/ml in PBS, PK-CA707-40050, Takara, Japan) was injected intravenously to stain the nuclei of dead cells in living mice. After 15 min, mice were sacrificed and PBS was perfused via the portal vein to drain the blood that contained excess EthD-3. Then the liver was processed using the 2D staining protocol described above.

## Statistics

In all animal experiments, the samples represent biological replicates derived from different mouse individuals. Representative data were supported by at least three biological replicates. Detailed sample size was estimated by considering the means and variation data from preliminary experiments. No randomization or blinding process was performed.

The F-test was used to check the homoscedasticity of the data, and the Kolmogorov-Smirnov test to check whether the data follow a Gaussian distribution. Significance tests were performed as described in the legends to each figure using Prism software (Graph pad, San Diego, CA).

## Mathematical modeling and simulation

In order to reveal the cellular behavior that underlies biliary tissue growth and remodeling, we traced the fate (i.e., the clone size of the progeny) of each single cell in vivo, made a simple growth model, and simulated it by computational methods.

### Data acquisition by 3D imaging

To determine the exact number of cells in a clone originating from a single BEC, we had to acquire a detailed 3D image for the entire clone in liver tissues. In many studies, the number of cells in a colony has been calculated or estimated on the basis of data obtained from 2D sectioned images. For example, in a previous study in which a similar statistical method was employed to reveal the growth mode of the epidermis (*Driessens et al., 2012*), the number of cells in a clone (clone size) was estimated from 2D section images. This was because the clones formed in the epidermis had an ordered shape and the actual clone size was well correlated with the estimations that can be derived from 2D section images. In stark contrast, the biliary tree exhibits branching and diversified 3D structures, which become far more complex under the liver injury condition, so that it is practically difficult to estimate clone sizes accurately from 2D section images. Hence, we chose to perform 3D imaging followed by direct cell counting to quantify the exact cell number in each colony. This approach is more time consuming than those relying on 2D image analyses, but it can reduce potential experimental errors and artifacts that might otherwise occur when calculating or estimating clone size.

In the quantitative single-cell tracing experiments, we analyzed the liver samples by making thick (300 µm) sections. This enabled us to observe the entire structure of the labeled clones in the biliary tree in 3D. We genetically labeled BECs at a very low frequency to perform single-cell tracing. This resulted in extremely low density (or, incidence) of labeled clones in the liver tissues, so that it was necessary to examine and observe vast numbers of samples in order for us to achieve data with enough size. One section of a 300 µm thickness has an equivalent data size to 30 sheets of 10-µm-

thick 2D sections. Thus, the 3D imaging not only provided the 3D structural information on clone sizes, but also enabled us to survey large areas in tissue samples very efficiently.

As mentioned before, the liver samples were treated with a nucleotide-staining reagent SYTOX Green. By counting the number of nuclei in a tdTomato$^+$ colony, we quantified the number of cells in the colony. It typically took only 10 hr to stain all of the nuclei in thick liver sections with SYTOX Green. It took a further 3 hr to treat the stained tissue with the clearing reagent SeeDB. We were thus able to complete these sample preparation steps within a day, showing that our protocol is suited for 3D observation of large quantities of tissue samples.

## Cell status and cell fate

Based on the distribution pattern of the clone-size data (*Figure 4*), we hypothesized the existence of heterogeneity in terms of proliferative capacity within BECs. We also presumed that BECs may take two different states, namely the proliferating and quiescent states, and tested this possibility by a nucleotide-incorporation experiment. We speculated that irreversible cell fate changes (in which BECs can change from the proliferating state to the quiescent state but not from the quiescent state to the proliferating state) took place, and the experimental results supported this hypothesis (*Figure 6b,e*).

The experimental results in *Figure 6* were consistent with the model we had proposed (*Figure 6b*, upper panel), but there was one point that did not completely match with our initial expectation. In theory, if our speculation represents reality and the experiment worked perfectly, the percentage of the BrdU and EdU double-positive cells in the EdU-positive population should reach 100%. In practice, however, it was approximately 80% (*Figure 6d*). A plausible explanation for this observed discrepancy is that the proportion is lower than the theoretical value (100%) when the 1st labeling fails to label proliferative cells completely, even when BrdU is administered continuously. We assumed that this was probably the case, and thus that the lower level of labeling might be explained by the stochastic characteristics of cell growth. We administered BrdU continuously for 8 days to achieve the maximum level of the 1st labeling so that the entire population of the proliferating cells could be labeled. According to the results of simulation, the cell division rate (denoted as 'm'), corresponding to the probability that any given cell will enter the cell cycle during a day, was relatively low (m = 0.175). In our stochastic model, the timing of the cell cycle entry per day was defined as a Bernoulli process. In this scenario, the final labeling rate of the 1st labeling was expected to be approximately 78.5% (if m = 0.175, then 1-(1-m)$^8$ = 0.78539). Thus, the observed discrepancy between the experimental data and the theoretical value for the proportion of the BrdU and EdU double-positive cells was brought about inherently and could be explained by the proliferative behavior of BECs. Our stochastic model and the quantitative data in *Figure 6* are mutually consistent with each other.

## Modeling and parameters

We simulated biliary epithelial tissue growth using computational Markov chain Monte Carlo methods. In the quantitative single-cell tracing experiments, the tissue growth up to 8 weeks of TAA injury was analyzed. We thus sought to reproduce the cellular behavior observed over this 8-week period by simulation. We defined the start of BEC proliferation to occur in a probabilistic manner, presuming that the average cell cycle duration (the period from the beginning of a cell division to the beginning of the next subsequent division) was longer than 24 hr. We defined a simulation model in which each proliferating cell could choose whether or not to start cell division, based on the cell division rate parameter 'm'.

When a cell enters the cell cycle, it should follow one of three fate options (*Figure 7a*). The probabilities of selection of each of these fates are defined by two parameters, 'r' and 's'. The parameter r affects stability of the colony size, while the parameter s represents imbalance between the selection of proliferative and quiescent states. In the case where totally balanced tissue growth occurs, such as in the normal maintenance of adult epidermis, the proportions of proliferating and quiescent cells are maintained and the parameter s can be ignored (s = 0) (*Doupé et al., 2010*). By contrast, the parameter s should be considered in biliary epithelial tissue growth because the biliary tree is expanding drastically (*Figure 2* and *Figure 2—figure supplement 2*) and the total cell numbers were not maintained during the process. We also added another parameter, 'p', which represented

the initial ratio of proliferating cells, as we found that there was a majority of quiescent cells in the BEC population (*Figures 4* and *6*).

In general, a negative effect, such as the density effect, should be included in the simulation when a model for growth in a limited space is to be constructed. In our simulation, such a negative effect, which could potentially be caused by an overgrowth of BECs, was not incorporated. It is true that there must be a limit to BECs' growth because they form a simple epithelial tissue and can only reside and expand within the liver of a certain size. However, within the period of 8 weeks of TAA injury, there still remained an empty space for BECs to grow and expand, and we observed that BECs continued to proliferate during the entire time course (*Figure 2—figure supplement 2*). Hence, we chose not to designate any parameter defining a density effect into our simulation. Such a parameter may become necessary if we are to simulate the growth of BECs at later time points, for example during the process of tumor formation.

## Code availability

The program code is available at the following address.

http://www.iam.u-tokyo.ac.jp/cytokine/research/simulation_code_ver_201.txt

## Acknowledgements

We thank N Miyata for cell sorting, N Imaizumi for animal care, and the members of the Miyajima lab for helpful discussions and advice. We thank Prof. Ian Alexander (Children's Medical Research Institute, Australia) for the pAM-LSP1-EGFP plasmid, Prof. R Jude Samulski and the NGVB Biorepository (University of North Carolina at Chapel Hill, USA) for the XX6-80 plasmid, Prof. Rolf Zeller for the pDIRE plasmid, and Penn Vector Core (University of Pennsylvania, USA) for the p5E18-VD2/8 plasmid.

## Additional information

### Funding

| Funder | Grant reference number | Author |
| --- | --- | --- |
| Japan Society for the Promotion of Science | Research Fellowship for Young Scientists | Kenji Kamimoto Kota Kaneko |
| Japan Society for the Promotion of Science | Postdoctoral Fellowship for Overseas Researchers | Cindy Yuet-Yin Kok |
| Core Research for Evolutional Science and Technology, Japan Science and Technology Agency | | Atsushi Miyajima |
| Japan Society for the Promotion of Science | KAKENHI, 22118006 | Atsushi Miyajima |
| Japan Society for the Promotion of Science | KAKENHI, 26253023 | Atsushi Miyajima |
| Japan Society for the Promotion of Science | KAKENHI, 24112507 | Tohru Itoh |
| Japan Society for the Promotion of Science | KAKENHI, 24590342 | Tohru Itoh |
| Japan Society for the Promotion of Science | KAKENHI, 26112704 | Tohru Itoh |
| Japan Society for the Promotion of Science | KAKENHI, 15H01369 | Tohru Itoh |
| Takeda Science Foundation | | Tohru Itoh |

The funders had no role in study design, data collection and interpretation, or the decision to submit the work for publication.

## Author contributions

KKam, Conception and design, Acquisition of data, Analysis and interpretation of data, Drafting or revising the article, Contributed unpublished essential data or reagents; KKan, Acquisition of data, Analysis and interpretation of data; CY-YK, Acquisition of data, Drafting or revising the article; HO, Acquisition of data, Analysis and interpretation of data, Contributed unpublished essential data or reagents; AM, Analysis and interpretation of data, Drafting or revising the article; TI, Conception and design, Analysis and interpretation of data, Drafting or revising the article

## Author ORCIDs

Tohru Itoh, http://orcid.org/0000-0002-6579-1638

## Ethics

Animal experimentation: All animal experiments were conducted in accordance with the Guideline for the Care and Use of Laboratory Animals of The University of Tokyo, under the approval of the Institutional Animal Care and Use Committee of Institute of Molecular and Cellular Biosciences, The University of Tokyo (approval numbers 2501, 2501-1, 2609 and 2706). Every effort was made to minimize animal suffering and discomfort and to reduce the number of animals used.

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
