## [Decision Letter]

Thank you for submitting your article "Heterogeneity and stochastic growth regulation of biliary epithelial cells dictate dynamic epithelial tissue remodeling" for consideration by *eLife*. Your article has been reviewed by two peer reviewers, and the evaluation has been overseen by Fiona Watt as Reviewing and Senior Editor.

The reviewers have discussed the reviews with one another and the Reviewing Editor has drafted this decision to help you prepare a revised submission. There are three major points that we wish you to address.

The first major point concerns the claim that the injury model is exclusively examining proliferation of BECs. The authors observe that there is no decrease in the Prom1 labeling index in the ductal reactions following TAA treatment (as compared to the initial index). They conclude from this that almost all of the BECs that arise during the ductal reaction come from pre-existing BECs. This would make TAA quite distinct from other injury models, such as DDC, carbon tetrachloride and CDE, in which a sizeable fraction of the ductal reaction cells are hepatocyte derived as shown in recent studies. In support of this claim, the authors report that they have done lineage tracing of hepatocytes using an adeno-associated virus – data that are being prepared for publication separately.

Because the unique features of this injury model are surprising and are important for the authors' conclusion that there is heterogeneity within the biliary compartment, it would provide additional confidence if this point were more carefully confirmed. For example, it is important to know that the system would be able to detect hepatocyte transdifferentiation into the biliary compartment if it were occurring. Hence, we recommend applying the Prom1 labeling system to an injury model associated with hepatocyte transdifferentiation (e.g. DDC) to ensure that a diminution in labeling index is seen – without this experiment, there is uncertainty about the claim that all of the biliary cells in the TAA model come from pre-existing BECs. It would also be helpful to supply the adeno-associated virus data to the reviewers. The presence of transdifferentiated BECs in the TAA ductal reaction would not negate the authors' conclusions, but it would represent a confounder that needs to be clarified.

The second major point pertains to the mathematical modeling, some of which supports the existence of a more proliferative subpopulation and some of which does not argue either way. The dual labeling experiments in Figure 5 suggest that BECs do indeed exist in one of two states – proliferative vs. non-proliferative. On the other hand, the fact that the distribution of colony size in Figure 3 is non-Gaussian does not rule out the possibility that statistical variation could account for the presence of the large colonies. In terms of the observation that proliferating cells form larger colonies in the ductular than in the ductal compartment, have the authors envisaged the possibility that the length of the cell cycle differs in duct and ductule cells? In addition, the irreversible switch to a quiescent state does not seem to fit with the notion proposed by Spencer et al., namely that the balance between quiescence and proliferation is controlled after each cell cycle at the end of mitosis (Spencer et al. Cell, 155(2):369-832013). Could the authors discuss this issue?

Third, the Discussion is confusing when it comes to the distinction between stochastic and deterministic features of the system. The dual labeling experiments indicate that there are proliferative and non-proliferative subsets of BECs, suggesting a deterministic (intrinsic) property. However, if a few "lucky" clones have proliferated more to give rise to the large colonies in Figure 3, this would suggest a stochastic (random variation) model of expansion. This section needs to be substantially re-written to be understandable by a general reader, particularly if the term stochastic growth regulation is to remain in the title.

---

## [Author Response]

There are three major points that we wish you to address.

The first major point concerns the claim that the injury model is exclusively examining proliferation of BECs. The authors observe that there is no decrease in the Prom1 labeling index in the ductal reactions following TAA treatment (as compared to the initial index). They conclude from this that almost all of the BECs that arise during the ductal reaction come from pre-existing BECs. This would make TAA quite distinct from other injury models, such as DDC, carbon tetrachloride and CDE, in which a sizeable fraction of the ductal reaction cells are hepatocyte derived as shown in recent studies. In support of this claim, the authors report that they have done lineage tracing of hepatocytes using an adeno-associated virus – data that are being prepared for publication separately.

Because the unique features of this injury model are surprising and are important for the authors' conclusion that there is heterogeneity within the biliary compartment, it would provide additional confidence if this point were more carefully confirmed. For example, it is important to know that the system would be able to detect hepatocyte transdifferentiation into the biliary compartment if it were occurring. Hence, we recommend applying the Prom1 labeling system to an injury model associated with hepatocyte transdifferentiation (e.g. DDC) to ensure that a diminution in labeling index is seen – without this experiment, there is uncertainty about the claim that all of the biliary cells in the TAA model come from pre-existing BECs. It would also be helpful to supply the adeno-associated virus data to the reviewers. The presence of transdifferentiated BECs in the TAA ductal reaction would not negate the authors' conclusions, but it would represent a confounder that needs to be clarified.

We agree that it is very important to clarify the contribution of hepatocytes to ductular reaction. In response to this critical comment, we have decided to include in the revised manuscript the hepatocyte lineage tracing data. Briefly we labeled hepatocytes with rAAV2/8-iCre in R26R-tdTomato mice, which were then applied to DDC or TAA liver injury model. The data is shown in the new Figure 3, with previous Figure 3–Figure 6 in the original version re-numbered as new Figure 4–Figure 7 in the revised version.

From the hepatocyte lineage tracing data, we confirmed that a large number of hepatocytes began to express some biliary markers (Sox9, Spp1, and MIC1-1C3) in DDC model, while only a very small number of hepatocytes that expressed other biliary markers, CK19, EpCAM and Prom1. This result fits well with recent reports about the conversion of hepatocytes into biliary-like cells (Tanimizu et al., J. Biol. Chem., 2014; Tarlow et al., Cell Stem Cell, 2014). Here, we use the terms “hepPDs” and “bilPDs” that were originally coined by Tarlow et al. (Cell Stem Cell, 2014). They reported that hepatocyte-derived biliary-like cells, hepPDs, were able to be characterized as Sox9^+^, Spp1^+^, MIC1-1C3^+^, but EpCAM^-^, CK19^-^, Prom1^-^. They also reported that pre-existing biliary cells that expand upon liver injury, bilPDs, are Sox9^+^, Spp1^+^, MIC1-1C3^+^, and EpCAM^+^, CK19^+^, Prom1^+^. We re-analyzed and summarized their data on gene expression profiles in hepPDs, bilPDs and hepatocytes (GSE5552) in an additional figure for the referees (Figure 8). The data clearly indicates phenotypic difference between hepPDs and bilPDs in terms of expression patterns of the aforementioned biliary markers. Importantly, our intent is not to deny the conversion (or trans-differentiation) of hepatocytes into biliary lineage, but to reiterate that previous reports and our results show that hepatocyte-derived biliary cells (hepPDs) are different from bilPDs derived from pre-existing biliary cells. Tarlow et al. showed hepPDs are different from bilPDs in terms of the ontogeny and gene expression profiles, and our data further supported this by showing that these populations were different from each other in terms of 3D structures (Figure 3). This means we can (and should) analyze the proliferative capacity of biliary cells that are derived from pre-existing biliary cells separately from those of hepatocytes-derived cells.

Author response image 1.**DOI:**
http://dx.doi.org/10.7554/eLife.15034.022

While the results of the hepatocyte tracing experiments may serve as strong and more direct evidence supporting the authenticity of our experimental settings using TAA, in order to answer reviewers’ request, we nevertheless performed the suggested experiments that focus on labeling index with our BEC labeling system in DDC model. We have included data in Figure 3—figure supplement 2in the revised manuscript. Mice were administered tamoxifen (10 mg/mouse). In that condition, around 30% of EpCAM^+^ BECs were labeled by the tdTomato reporter, and around 16% of MIC1-1C3^+^ cells were labeled. Then mice were fed with DDC diet for 8 weeks. FACS analyses were performed to quantify the labeling rate of BECs before/after liver injury with two different BEC markers, EpCAM and MIC1-1C3. As expected, we did not find a decrease in labeling rate in EpCAM+ populations after DDC administration. We found tendency for the labeling rate in MIC1-1C3+ populations to be reduced, but were unable to confirm whether the observed difference was significant (N=2 for the normal and DDC conditions, respectively). The percentage of the MIC1-1C3^+^ tdTomato^+^ populations was: 16.4% and 15.7% for the normal condition (N=2); and, 14.4% and 15.3% for the DDC condition (N=2). Unfortunately, we did not have any more mice that were available for the analysis during the two-month period allowed for revision. Considering the relatively small contribution of the hepatocyte-derived cells to the entire MIC1-1C3^+^ population upon DDC injury (1.88%, shown in Figure 3—figure supplement 1 in the revised manuscript), the result is overall consistent with that of the hepatocyte lineage tracing data because most of hepatocyte-derived biliary cells (hepPDs) are MIC1-1C3^+^ and EpCAM^-^.

In general, the studies about hepatocyte-biliary conversion may sometimes appear to be complicated and controversial. Nevertheless, we can explain our data and recent other reports logically and consistently when we carefully consider the expression profiles of various types of biliary epithelial cell markers. In the present study, we are focusing on bilPDs, rather than hepPDs, that form the main structural framework for ductular reaction.

The second major point pertains to the mathematical modeling, some of which supports the existence of a more proliferative subpopulation and some of which does not argue either way.

We agree that this point is really important for this study and apologize that our initial manuscript was not sufficiently clear. We have revised our description related to dual labeling experiment, single cell tracing, and simulation. Nevertheless, we think that the results of single cell tracing and mathematical model do not contradict the existence of proliferative state that was shown by dual labeling experiments.

In the dual labeling experiment (Figure 6 in the revised manuscript), we suggested that BECs could be divided into two types, proliferating cells and non-proliferating cells. The definition was based on statistic relationships of “two cell divisions” (1st labeling the and 2nd labeling). Thus, the term proliferative or non-proliferative refer to temporal state of cell division (two rounds of cell cycle), and we did not define the state, proliferative or quiescent, to be perpetually proliferating. From this experiment, we cannot make assumptions about the whole history of cell fate (that may contain large number of cell division) during colony formation in chronic liver injury periods by TAA. And importantly, there exists the possibility that proliferative cells could be changed into the non-proliferative state, though we do not know when this may occur. Thus, existence of proliferative state does not refute the stochastic model, and existence of temporal proliferative state does not simply mean the deterministic model. In order to generate a model that considers the whole history of cell divisions, we had to perform simulation analysis (Figure 7 in the revised manuscript).

The dual labeling experiments in Figure 5 suggest that BECs do indeed exist in one of two states – proliferative vs. non-proliferative. On the other hand, the fact that the distribution of colony size in Figure 3 is non-Gaussian does not rule out the possibility that statistical variation could account for the presence of the large colonies.

We apologize about this point and thank the referees for this critical comment. In our original manuscript, we wrote that non-Gaussian distribution means that the large colonies did not result from statistical variation. We admit, however, that it was an overstatement because some statistical variation can indeed make non-Gaussian distributions. We appreciate that the comment made our manuscript accurate. We have deleted that particular sentence in the revised manuscript, while the overall conclusion and concept of our study was not changed or affected.

*In terms of the observation that proliferating cells form larger colonies in the ductular than in the ductal compartment, have the authors envisaged the possibility that the length of the cell cycle differs in duct and ductule cells?*

We found that most BECs in the ductal compartment remained as single or two cells after 8 weeks of liver injury. In addition, from Ki67 staining experiment (Figure 5), we found almost no Ki67^+^ cells in the ductal compartment. Thus we thought that BECs in ductal compartment remain quiescent (G0 phase) rather than undergoing a long cell cycle, because Ki67 is generally expressed in G1, S, G2 and M phases. Therefore, we believe the difference of colony size between ductular compartment and duct compartment was not due to difference of cell cycle length.

In mathematical simulation, we defined quiescent cells as those that do not divide at all. This means we do not need to consider their cell cycle length. On the other hand, we had to consider the cell cycle length of proliferative cells. We first ran many simulations with uniform cell cycle length for all cells. However, we found that the best fit was achieved when we allowed for variation of cell cycle length among the cells. The variation was defined as the parameter m (See Supplementary Methods for details). Interestingly, this result agrees with the notion proposed by Spencer et al. (Spencer et al. Cell, 155(2):369-383, 2013; which was suggested by the reviewers in the comment below). Briefly, the study confirmed by their single cell experimental system that there is a large cell-to-cell variation of cell cycles and delay between mitogen re-stimulation and CDK activation. They also reported that the cells had heterogeneous inter-mitotic times ranging from 20 hr to >50 hr. This fits well with our stochastic proliferation model, because we defined our model to have various inter-mitotic times.

In addition, the irreversible switch to a quiescent state does not seem to fit with the notion proposed by Spencer et al., namely that the balance between quiescence and proliferation is controlled after each cell cycle at the end of mitosis (Spencer et al. Cell, 155(2):369-832013). Could the authors discuss this issue?

We appreciate notifying us an important and insightful report about cell proliferation. We have added some discussion referring to it in the revised manuscript. It is true that there is some discrepancy, as well as common points. We think the discrepancy may be due to the role of BECs in vivo. During tissue remodeling and tissue growth process of biliary epithelial cells, BECs must form a functional tubular structure. We assume that BECs must gain a sort of steady-state phenotype (or, become more matured and differentiated cells) in order to generate and maintain a functional and robust epithelial tubular structure. From this point of view, it is reasonable to assume that proliferation state changes irreversibly in vivo. Unfortunately, there is no established marker so far that can clearly distinguish different stages of BECs in the course of their functional differentiation and maturation, which prevented us from directly evaluating this possibility. Nevertheless, the assumption could be supported partially by our findings that revealed the relationship between the proliferative capacity of BECs and structural feature of the biliary tree (Figure 4), as BECs constituting different tissue structures may be in distinct differentiation stages (Figure 4 in the revised manuscript). We have included discussion on this point in the revised manuscript.

Third, the Discussion is confusing when it comes to the distinction between stochastic and deterministic features of the system. The dual labeling experiments indicate that there are proliferative and non-proliferative subsets of BECs, suggesting a deterministic (intrinsic) property. However, if a few "lucky" clones have proliferated more to give rise to the large colonies in Figure 3, this would suggest a stochastic (random variation) model of expansion. This section needs to be substantially re-written to be understandable by a general reader, particularly if the term stochastic growth regulation is to remain in the title.

We deeply apologize that our Discussion appeared to be confusing and our definition of several words (proliferative, stochastic, and deterministic) might be obscure. We have re-written the Results and Discussion parts to be more clear, precise and understandable. We have also added clear definitions of words, especially conceptual words. We hope the re-defining and words revision make our concept clearer. Basically, our word definition is the same as that of a previous report (Doupe, et al., 2010). We showed the existence of proliferative and non-proliferative states in BECs, but this does not mean that they are deterministic. Here, deterministic refers to restriction of the cell to a state that is permanently fixed, while our finding about proliferative state is referring to temporal state, not to the eventual cell fate. In our model, “switching” of their proliferative state is important.

The concept of stochastic model does not refute the existence of a proliferative sub population. Cells can be classified into two distinct proliferative states, which can switch in a stochastic manner. We had chosen the concept of “stochastic” for our model in order to explain the heterogeneity of proliferative capacity (shown in Figure 4 in the revised version) accounting for the existence of transient proliferative sub states (shown in Figure 6 in the revised version). In other words, the stochastic feature of biliary cell proliferation was necessary (and sufficient) to explain our data simultaneously. We have emended explanations in main manuscript about our mathematical modeling and feature of “stochastic cell behavior”. The above description in response to the second major point would also be the answer to solve the discrepancy between the results about dual labeling and colony size distributions.